# Revisiting Bi-Linear State Transitions in Recurrent Neural Networks

**M.Reza Ebrahimi**
Qualcomm AI Research
ebrahimi@qti.qualcomm.com

**Roland Memisevic**
Qualcomm AI Research [*]
rmemisev@qti.qualcomm.com

## Abstract

The role of hidden units in recurrent neural networks is typically seen as modeling memory, with research focusing on enhancing information retention through gating mechanisms. A less explored perspective views hidden units as active participants in the computation performed by the network, rather than passive memory stores. In this work, we revisit bilinear operations, which involve multiplicative interactions between hidden units and input embeddings. We demonstrate theoretically and empirically that they constitute a natural inductive bias for representing the evolution of hidden states in state tracking tasks. These are the simplest type of tasks that require hidden units to actively contribute to the behavior of the network. We also show that bilinear state updates form a natural hierarchy corresponding to state tracking tasks of increasing complexity, with popular linear recurrent networks such as Mamba residing at the lowest-complexity center of that hierarchy.

## 1 Introduction

State tracking is a fundamental requirement for performing sequential decision-making tasks, in which future actions depend on the consequences of past actions. The consequences of past actions are usually not directly observable, making state tracking a key ingredient in virtually every real-world multi-step interaction between an agent and its environment. This includes multi-hop dialogue, end-to-end learned robot control, and recent "agentic LLM" use-cases, in which a language model is trained to interact with an API.

While state tracking is an ill-defined concept in general, a common way to define it formally, which shall suffice for the purpose of this work, is to treat it as the task of correctly representing the arbitrary-length sequence of states that a state machine takes on in response to observing a given sequence of inputs. This is equivalent to modeling Finite Automata (FA), or regular languages, in the Chomsky hierarchy of formal languages (Chomsky, 1956; Hopcroft et al., 2006).

Although state tracking is a seemingly simple task for neural networks to learn, many models are surprisingly bad at learning it from data. The reason for its simplicity is that the task admits a simple inductive decomposition: For each input in the sequence, it suffices to update an internal representation of the state inferred from all inputs seen previously. As a result, it is possible, in principle, to learn state tracking for sequences of arbitrary length by simply learning the appropriate state transitions for every (input, state)-pair from the training data.

However, in practice, this requires an inductive bias towards the input-by-input state update, which is not present in many models. For example, many popular sequence models, such as the Transformer cannot learn to perform state tracking (Dziri et al., 2023; Anil et al., 2022; Abbe et al., 2024) on sequences longer than the training data. This includes very large, pre-trained Transformer-based

---

[*] Qualcomm AI Research is an initiative of Qualcomm Technologies, Inc.

39th Conference on Neural Information Processing Systems (NeurIPS 2025).

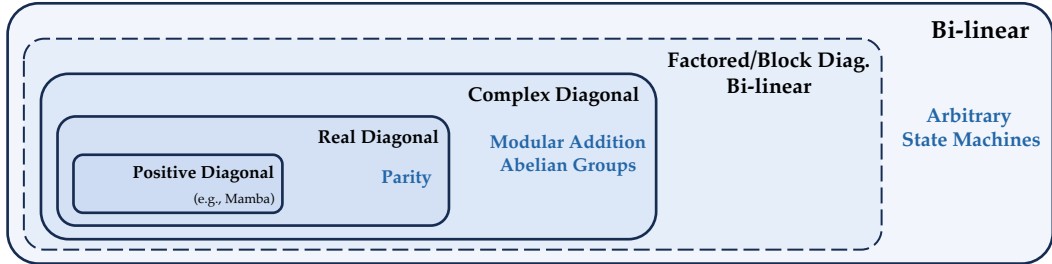

Figure 1: Taxonomy of bilinear RNNs studied in this paper, along with example regular language tasks they can learn (in blue).

language models, and it is the case even when trained to use chain-of-thought reasoning ("inference-time compute") (e.g., Ebrahimi et al. (2024)).

Similarly, as shown by Merrill et al. (2024), many linear recurrent networks fail to learn arbitrary-length state tracking tasks, which include large RNN-based pre-trained language models, such as Mamba (Gu and Dao, 2024), or the mLSTM (Beck et al., 2024). Recent work has shown that certain linear RNNs can learn some state tracking tasks, if the hidden-to-hidden transition matrix satisfies two conditions: (i) it is a function of the input (and thus not time-invariant), and (ii) not all of its eigenvalues are positive (Sarrof et al., 2024; Grazzi et al., 2025). However, the tasks that can be learned under these conditions are highly restricted as we shall show. A benefit of linear models, besides being amenable to analysis, is that they can be trained efficiently on parallel hardware (e.g., Martin and Cundy (2018)). This is in contrast to standard (non-linear) recurrent networks (RNNs), due to the linear dependence between hidden states across time-steps.

In this work, we revisit recurrent networks with *bilinear* hidden-to-hidden transitions. The transition matrix in these models is a simple bilinear function of inputs and hidden activations of the previous time-step. Various types of bilinear recurrent models have been investigated in the past (e.g., Sutskever et al. (2011); Downey et al. (2017); Wu et al. (2016)), but they have not caught on as widely used models. This is in part due to instabilities and optimization difficulties owed to their inherent three-way multiplicative interactions.

We show that bilinear RNNs are highly effective at learning state tracking tasks if one leverages a few simple tricks to avoid instabilities during training and inference. This includes removing any additive components ("bias terms" and other additive contributions to the hidden state), such that the hidden state is a true bilinear not an affine function of the previous time-step hidden state and input. We also show that bilinear models form a natural hierarchy of decreasing complexity, ranging from fully unconstrained but parameter-inefficient models to highly constrained and parameter-efficient models. The different model classes within the hierarchy correspond to increasingly narrow subclasses of regular language modeling tasks that can be learned from data (see Fig. 1). Several existing linear RNNs, such as Mamba (Gu and Dao, 2024), are at the center of the hierarchy, with no state tracking capability at all.

A task that has received significant attention as a testbed for learning state tracking behavior with sequence models in the past is the task of computing the parity of a binary bit string. We show that a notable special case of learning bilinear state transitions without additive terms is that it can learn the parity task with a frozen (untrained) recurrence and only training final readout layer on as few as two training examples.

Figure 1 shows an overview of bilinear models along with task classes we study in this work, ranging from simulating arbitrary state machines (the broadest class learnable by unconstrained bilinear models) to parity (the most narrow class, which can be learned even by models with real-valued diagonal transition matrix). Further constraining the transition matrix to positive diagonal impedes a model's ability to perform state tracking (see, e.g., Grazzi et al. (2025); Sarrof et al. (2024)). We summarize our contributions as follows:

- We revisit bilinear state transitions in RNNs and present an extensive study, showing that they can learn state tracking tasks, unlike many existing linear recurrent models, albeit with the caveat that they can have a very large number of parameters.

- We show that it is always sufficient (and in some cases necessary) for the hidden state to be a *pure linear not affine* function of the hidden state at the previous time-step. The absence of any additive terms makes hidden states scale-invariant, which in turn allows us to normalize hidden states during training and/or inference without sacrificing the linear recurrence.
- We show that a pure linear (not affine) RNN with frozen random weights and a trained linear readout layer can learn parity with perfect accuracy from only two training examples.
- We show that linear RNNs with diagonal transition matrices are a special case limited to learning state tracking tasks with commutative structure. This restriction is true even for complex-valued diagonal transition matrices. Hence, linear RNNs with block-diagonal transition matrices of size $2 \times 2$ are *not* able to learn general state machines (negative result).

**Related work:** Bilinear models have been studied extensively for unsupervised learning of transformations and relationships from data (Tenenbaum and Freeman, 1996; Olshausen et al., 2007; Memisevic and Hinton, 2010). Bilinear state transitions have also been discussed in the context of recurrent networks by Sutskever et al. (2011); Downey et al. (2017); Wu et al. (2016); Michalski et al. (2014). Besides the analysis, a key novelty in our practical results is the importance of using pure bilinear, not affine, state transitions. As a special case of bilinear state transitions, we study the use of two-dimensional subspaces in which hidden units are transformed through rotations only. This is similar to existing, but non-linear, networks with complex-valued or unitary transition matrices (e.g., Arjovsky et al. (2016); Wolter and Yao (2018)).

Recent work has shown that a dependence of hidden state transitions on the inputs is necessary for a recurrent network to learn *any* state tracking behavior (Grazzi et al. (2025); Sarrof et al. (2024); Fan et al. (2024)), although the connection to bilinear models is absent in that work, and transition matrices are defined as neural network layers and include input-dependent additive terms (which we show to be detrimental to learning). Terzić et al. (2025) propose a variant of state-space models where the transition matrix is constructed from an input-dependent linear combination of learned (but fixed) dense matrices, enabling some degree of length generalization on a set of regular language tasks. Bilinear models learn to encode hidden-to-hidden transitions as linear functions of the input, making them reminiscent of observable operator models and predictive state representations (Jaeger, 2000; Littman and Sutton, 2001).

## 2   Modeling hidden state dynamics using bilinear state transitions

A linear recurrent neural network represents a sequence of observations $x^t \in \mathbb{R}^D$ via the temporal evolution of a vector of hidden variables (the "hidden state") $h^t \in \mathbb{R}^H$. The most common form for modeling the temporal evolution is:

$$h^t = \mathcal{A}h^{t-1} + \mathcal{B}x^t + b, \tag{1}$$

where $\mathcal{A} \in \mathbb{R}^{H \times H}$ is a hidden-to-hidden matrix, $\mathcal{B} \in \mathbb{R}^{H \times D}$ is an input-to-hidden matrix modeling input-dependent additive terms, and $b \in \mathbb{R}^H$ is a vector of additive input-independent biases.

Recently, it has been remarked that for a recurrent network of the form Eq. (1) to be able to learn state tracking tasks, the hidden-to-hidden transformation $\mathcal{A}$ needs to depend on the input $x$ (Gu and Dao, 2024; Sarrof et al., 2024; Fan et al., 2024; Grazzi et al., 2025). The necessity for input-dependence has been motivated by showing, both theoretically and empirically, that models fail to learn state tracking tasks in the absence of the input-dependence. The exact form of the dependence of $\mathcal{A}$ on $x$ has been left open. Instead, it has been suggested to parameterize $\mathcal{A}(x)$ as a neural network.

We argue that a natural alternative for this dependence, while keeping the recurrence linear, is to make it multiplicative, such that a hidden unit $h_i^t$ at time $t$ is a function of the products $h_j^{t-1} \cdot x_k^t$ of the components of the hidden state at the previous time-step and the inputs at the current time-step. The reason is that this makes explicit the input-dependent transformations between hidden states across time-steps, and thereby makes it natural to simulate a state machine, as we shall discuss below.

### 2.1   Simulating finite-state machines and group structures

A finite-state machine (FSM), or finite automaton (FA), can be formally defined as a tuple $\mathcal{S} = (Q, \Sigma, \delta, q_0)$, where $Q$ is a finite set of states, $\Sigma$ is a finite input alphabet, and $\delta : Q \times \Sigma \to Q$ is

the state transition function. The machine starts in an initial state $q_0 \in Q$. Given an input sequence $\sigma = \{\sigma_1, \sigma_2, \ldots, \sigma_T\}$, the FSM undergoes a sequence of state transitions $\{q_1, q_2, \ldots, q_T\}$, where each state $q_t$ is determined by $q_t = \delta(q_{t-1}, \sigma_t)$. We shall define state tracking formally as the task of simulating a state machine. A model is said to simulate state machine $\mathcal{S}$ if, after observing the complete input sequence $\sigma$, it can produce the final state $q_T$ (see Deletang et al. (2023); Liu et al. (2023)).

As a special case, we can consider state machines representing a group structure. In this context, a group $(G, \cdot)$, where $\cdot$ denotes the group operation, can be modeled as an FSM where the set of states and the input alphabet are identical to the set of group elements, i.e., $\Sigma = Q = G$. The transition function is defined by the group operation itself: $\delta(g_1, g_2) = g_1 \cdot g_2$ for all $g_1, g_2 \in G$, representing the associative group operator with corresponding inverse and identity group elements.

Another important special case is that of an abelian group, where the group operator is commutative. We consider integer groups under addition modulo $m$, denoted as $\mathbb{Z}_m$. In this case, the state set and input alphabet are $Q = \Sigma = \mathbb{Z}_m = \{0, 1, \ldots, m-1\}$. The transition function $\delta : \mathbb{Z}_m \times \mathbb{Z}_m \to \mathbb{Z}_m$ is defined by addition modulo $m$: $\delta(a, b) = (a + b) \pmod{m}$, for all $a, b \in \mathbb{Z}_m$. Specifically, simulating the group $\mathbb{Z}_2$ is equivalent to computing the parity of a binary input sequence. It is important to note that the operation of integer addition modulo $m$ is the canonical commutative operation to consider, as all finite abelian groups are structurally similar (isomorphic) to direct products of subgroups of $\mathbb{Z}_m$.

## 2.2 Bilinear RNNs can learn arbitrary state machines

Formally, we consider the hidden state $h^t$ to be a bilinear function of the previous hidden state $h^{t-1}$ and the current input $x^t$. As such, we model state transitions as:

$$h_i^t = (h^{t-1})^\top \mathcal{W}_i \, x^t = \sum_{jk} \mathcal{W}_{ijk} x_k^t h_j^{t-1}, \tag{2}$$

where $\mathcal{W}_{ijk}$ are the components of a three-way parameter tensor $\mathcal{W} \in \mathbb{R}^{H \times H \times D}$, with matrix $\mathcal{W}_i \in \mathbb{R}^{H \times D}$ denoting the $i$-th slice of the tensor. Note that Eq. (2) is equivalent to using an input-dependent transition matrix $\mathcal{A}_x$ such that:

$$h^t = \mathcal{A}_x h^{t-1}, \tag{3}$$

with $(\mathcal{A}_x)_{ij} = \sum_k \mathcal{W}_{ijk} x_k$. In other words, the state transition matrix $\mathcal{A}$ is fully parameterized through a linear transformation of the input $x$. RNNs with bilinear state transitions (albeit typically in affine, not pure multiplicative form) have been studied previously (e.g., Sutskever et al. (2011); Downey et al. (2017); Wu et al. (2016)).

We note that the multiplicative dependence, in particular in the absence of any additive contributions from the input, allows the inputs to "route" the information flow in the hidden states, or conversely represent transformations acting on them. The ability for layers in a network to elicit transformations acting on other layers has been a common motivation for studying trainable bilinear models in the past (e.g.,Olshausen et al. (2007)). In the context of Eq.(2), it allows inputs to determine the temporal evolution of the hidden state (or elicit "computations" to be performed by the hidden layer), which is different from the somewhat more common view of the role of RNN hidden units as memorizing information. In the following we shall now make this perspective more concrete by showing that a bilinear RNN can simulate any state machine.

**Proposition 1.** *The bilinear state transition model defined in Equation (2) is capable of simulating any finite state machine $\mathcal{S} = (Q, \Sigma, \delta, q_0)$.*

We refer to Appendix A.1 for the proof. Besides allowing inputs to encode transformations on hidden units, the absence of any additive terms in Eq. (2) makes the hidden units scale-invariant. In other words, (up to floating point accuracy) one can multiply a hidden state vector by a constant at any time-step and divide by the same constant at a later time-step without any effect on the final result. In Section 3 we shall show that the scale-invariance allows us to keep hidden activations stable during training and inference.

### 2.2.1 Factorized state transition tensor

The bilinear state update in Eq. (2) utilizes a three-way parameter tensor $\mathcal{W}$, whose $H^2 D$ parameters often necessitate low-rank factorization for a more parsimonious model. One of the most common

factorization methods proposed in the literature is the Canonical Polyadic (CP) decomposition, also known as Parallel Factor Analysis (PARAFAC) (Hitchcock, 1927), which was used, for example, in the bilinear RNNs discussed in Sutskever et al. (2011); Downey et al. (2017). The CP decomposition approximates the tensor as a sum of $R$ rank-1 tensors $\mathcal{W} = \sum_{r=1}^{R} w_r^{(h_1)} \otimes w_r^{(h_2)} \otimes w_r^{(x)}$, which in terms of individual components, is expressed as: $\mathcal{W}_{ijk} = \sum_{r=1}^{R} \mathcal{W}_{ir}^{(h_1)} \mathcal{W}_{jr}^{(h_2)} \mathcal{W}_{kr}^{(x)}$. Here, $\otimes$ denotes the outer product, and vectors $w_r^{(h_1)}$, $w_r^{(h_2)}$, and $w_r^{(x)}$ are the component vectors for the $r$-th rank-1 term. These component vectors are collected as columns in the factor matrices $\mathcal{W}^{(h_1)} \in \mathbb{R}^{H \times R}$, $\mathcal{W}^{(h_2)} \in \mathbb{R}^{H \times R}$, and $\mathcal{W}^{(x)} \in \mathbb{R}^{D \times R}$, respectively. As a result, the total number of parameters is reduced to $R(2H + D)$. The input-dependent state transition matrix $\mathcal{A}_x$ can then be expressed compactly as:

$$\mathcal{A}_x = \mathcal{W}^{(h_1)} \mathrm{diag}\left((\mathcal{W}^{(x)})^\top x\right)(\mathcal{W}^{(h_2)})^\top, \tag{4}$$

where input vector $x \in \mathbb{R}^D$ and $\mathrm{diag}(\cdot)$ constructs a diagonal matrix from a vector. As shown in Appendix C.3, an increasing number of factors ($R$), enables simulating state machines with larger states, as factored models with a larger $R$ provide a better approximation of a full bilinear model.

### 2.2.2 Block-diagonal state transition tensor

An alternative method for controlling the parameter count in the bilinear model involves imposing block structures on the effective state transition matrix $\mathcal{A}_x$. This is achieved by utilizing $B$ distinct three-way parameter tensors, denoted as $\mathcal{W}^{(b)} \in \mathbb{R}^{H' \times H' \times D}$, one for each block $b \in \{1, \ldots, B\}$. Here, $H' = H/B$ represents the dimensionality of each block's corresponding state subspace, assuming $H$ is an integer multiple of $B$.

Consequently, the overall state transition matrix $\mathcal{A}_x$ adopts a block-diagonal structure, where each diagonal block, $\mathcal{A}_x^{(b)} \in \mathbb{R}^{H' \times H'}$, is generated from its respective tensor $\mathcal{W}^{(b)}$ via the relation $(\mathcal{A}_x^{(b)})_{ij} = \sum_k \mathcal{W}_{ijk}^{(b)} x_k$. As a result, the total number of parameters is reduced by a factor $B$. This block-diagonal parameterization can be conceptualized as employing $B$ independent "heads", each processing a distinct subspace of the hidden state vector using its own dense transition dynamics. It is reminiscent of block-diagonal transition matrices studied by Fan et al. (2024), albeit defining them as a bilinear, non-additive function of the inputs.

### 2.3 Complex diagonal bilinear RNNs

In Section 2.2, we established that the bilinear state-transition form defines an input-dependent state-transition matrix $\mathcal{A}_x$, whose entries are linear functions of the input $x$, parameterized by a third-order tensor $\mathcal{W}$. In this section, we discuss how diagonalizing the state-transition matrix, a common simplification in many linear RNN variants, reduces a model's expressive capability to commutative operations.

First, consistent with common practice in the linear RNN literature (e.g., Orvieto et al. (2023)), we consider state-transition matrices that are diagonalizable over the complex numbers. This focus is justified because the set of non-diagonalizable matrices has measure zero (Axler, 2024); consequently, any matrix $A \in \mathbb{R}^{N \times N}$ can be made diagonalizable over $\mathbb{C}$ through an arbitrarily small perturbation of its entries (Zhinan, 2002). This implies that, based on the real Jordan Normal Form, the state-transition matrix $\mathcal{A}_x$ can be expressed as:

$$\mathcal{A}_x = \mathcal{P}_x \mathcal{D}_x \mathcal{P}_x^{-1}, \tag{5}$$

where $\mathcal{P}_x \in \mathbb{R}^{H \times H}$ is an invertible matrix and $\mathcal{D}_x \in \mathbb{R}^{H \times H}$ is a real block-diagonal matrix with blocks of size $1 \times 1$ (for real eigenvalues) or $2 \times 2$ of the form $\mathcal{C}_2 = \begin{pmatrix} a & -b \\ b & a \end{pmatrix}$ (for complex conjugate pairs of eigenvalues $a + ib$). Both $\mathcal{P}_x$ and $\mathcal{D}_x$ are generally parameterized by the input $x$.

A particularly important special case is when the state-transition matrix $\mathcal{A}_x$ is orthogonal. This is highly desirable for linear RNNs, as it ensures stability by guaranteeing that all eigenvalues of $\mathcal{A}_x$ have unit norm, preventing exploding or vanishing states during recurrent updates. In this scenario, the diagonal matrix $\mathcal{D}_x$ will be entirely composed of 2-dimensional rotation matrices, which we denote as $\mathcal{R}_2 = \begin{pmatrix} \cos\theta & -\sin\theta \\ \sin\theta & \cos\theta \end{pmatrix}$.

A common simplification is to assume that the transformation matrix $\mathcal{P}_x$ is independent of the input $x$ (i.e., $\mathcal{P}_x = \mathcal{P}$). This fixed matrix $\mathcal{P}$ can then be canceled out in the recurrence steps and absorbed into the input and output transformations of the recurrent layer. The state dynamics are thus governed by $\mathcal{A}_x = \mathcal{D}_x$, which remains input-dependent and retains its block-diagonal structure, often simplified to purely diagonal with real, or even non-negative entries, e.g., in Mamba (Gu and Dao, 2024). [2]

It is important to recognize that fixing $\mathcal{P}$ while $\mathcal{D}_x$ varies with the input implies that the overall state-transition matrices $\mathcal{A}_x = \mathcal{P}\mathcal{D}_x\mathcal{P}^{-1}$ and $\mathcal{A}_y = \mathcal{P}\mathcal{D}_y\mathcal{P}^{-1}$ for different inputs $x$ and $y$ will commute if and only if their corresponding block-diagonal components $\mathcal{D}_x$ and $\mathcal{D}_y$ commute. Thus this architectural choice inherently restricts the model to commutative transition dynamics (Terzić et al., 2025). In fact, a model whose transition matrix $\mathcal{A}_x$ is directly parameterized as such a block-diagonal matrix $\mathcal{A}_x = \mathcal{D}_x$ (i.e., effectively $\mathcal{P} = \mathcal{I}$) can naturally represent operations from *any* abelian group, as we show in the following proposition:

**Proposition 2.** *A linear RNN of the form in Eq.* (3) *with orthogonal state-transition matrices $\mathcal{A}_x$ that share a common, input-independent eigenbasis (i.e., $\mathcal{A}_x = \mathcal{P}\mathcal{D}_x\mathcal{P}^{-1}$ with fixed $\mathcal{P}$) can simulate any abelian group (commutative operation).*

We refer to Appendix A.2 for the proof. In Section 3.5, we also present a visualization of the invariant subspaces and rotation angles learned by the model.

## 2.4 Real diagonal bilinear RNNs

Finally, the block-diagonal transition matrix, $\mathcal{D}_x$, is often further simplified to be purely diagonal with real values; e.g., the RG-LRU cell utilized in the Hawk architecture (De et al., 2024). However, as noted by Grazzi et al. (2025), such models are incapable of learning modular addition.

Contrary to this limitation, we will show that learning parity is not only straightforward, but that it is in fact trivial for a linear RNN with real-valued diagonal state transitions which depend multiplicatively not additively on $x$. Length-generalization on the parity task is widely used to test the state tracking capabilities of sequence models (e.g., Anil et al. (2022); Grazzi et al. (2025)).

**Proposition 3.** *A random network with frozen real-diagonal transition matrix (without additive terms) and learnable linear readout layer learns parity with probability $1 - 2^{-H}$, for arbitrary sequence length from only 2 training examples of odd and even parity.*

Freezing the recurrent weights turns the network effectively into a bilinear variant of an echo state network (Jaeger, 2007; Maass et al., 2002). Our result shows that an echo state network with state transitions depending *only multiplicatively* on the input can learn parity. We shall show experimental results confirming this result in practice in Section 3.4. This is in contrast to models like Mamba (Gu and Dao, 2024), in which state transitions are diagonal and positive, and which can therefore not learn parity even when adapting recurrent parameters during learning (Grazzi et al., 2025).

# 3 Experiments

**Tasks:** To evaluate the state-tracking capabilities of the bilinear RNN model variants introduced previously, we use the following three tasks: modular addition, random state machine, and modular arithmetic. In the modular addition task, the model processes a sequence of integers, each randomly drawn from the set $\mathcal{Z}_m = \{0, \cdots, m-1\}$, and is required to predict their sum modulo $m$. For the random state machine task, the model must simulate a randomly generated finite-state machine where both the input alphabet $\Sigma$ and the set of states $Q$ are identical to $\mathcal{Z}_m$; and for each $q \in Q$, the transition function is set to $\delta(q, \sigma) = \pi_q(\sigma)$, where $\pi_q$ is a random permutation of $\Sigma$. Finally, the modular arithmetic task involves processing a sequence alternating between integers from $\mathcal{Z}_m$ and arithmetic operators (from the set $\{+, \times, -\}$); the model must compute and output the result of these operations, applied sequentially, with all calculations performed modulo $m$. For all tasks, multi-digit integers are tokenized into single tokens. We refer the reader to Appendix B.1 for examples and additional details on each task.

**Models:** We compare full, factored, and block-diagonal bilinear models against several baseline architectures: non-linear RNNs including LSTM (Hochreiter and Schmidhuber, 1997) and Elman

---

[2]Note, however, that this restriction may not be as problematic when additive terms are present.

RNN (Elman, 1990), Mamba, and Transformer models. All RNN-based models (including the bilinear, and non-linear ones) have a single recurrent layer followed by a linear classification head over the hidden states. For the Mamba and Transformer baselines, we evaluate configurations with 1, 2, and 4 layers. A consistent hidden and input dimensionality of 256 is used across all models. Further details of the experimental setup are provided in Appendix B.

**Recurrence stability:** The absence of additive terms in recurrent formulation of Eq.(2) makes the introduced bilinear hidden states scale-invariant as discussed above. For inference, we can therefore normalize hidden states during the recurrence, while not doing so during training, without introducing any inconsistency between training and inference.

## 3.1 Main results

We trained the models described above on the three tasks, on instances of lengths 2-10, and evaluated on instances of length 500. The training is done for 100,000 steps, where at each step 64 samples (of length 2-10) are generated for the task. Each model is trained using three different learning rates and picked the best-performing model. Table 1 summarizes the main results. In all tables, we scale accuracy values such that 0 represents random chance, and 1 is perfect accuracy.

We observe that bilinear models generally perform best across all tasks. Bilinear block-diagonal variants exhibit improved performance as the block size increases. Notably, the real-diagonal model (a bilinear block-diagonal model with block size 1) can only learn parity (i.e., modular addition with $m = 2$); however, increasing the block size to two enables the learning of modular addition for larger values of $m$, as demonstrated in Proposition 2. Also, the $\mathcal{R}_2$ block-diagonal model explicitly parametrizes the state transition matrix as rotation blocks of the form $\mathcal{R}_2$, with angles parameterized by inputs. In contrast, a block-diagonal bilinear with block size of 2 parametrizes 2D-blocks freely from the input.

Non-linear recurrent models, such as LSTM and simple RNN, also perform well on these state-tracking tasks. It can be speculated that multiplicative interactions between hidden states and inputs arise from the gating mechanisms and non-linear activation functions within their recurrent structures. While Mamba can learn the tasks in-distribution for small state sizes $m$, it largely fails to generalize to longer sequences. Also, the failure of transformers in length-generalization is a well-known observation in the literature.

Note that in Table 1, we used a consistent hidden dimension of 256 across all models. In Appendix C.1, we report the performance of models matched in parameter count by adjusting their hidden dimensions. In addition, Appendix C.2 presents results for simulating dihedral groups with various models.

## 3.2 Data Efficiency

We showed in the previous section that bilinear models are effective at learning state tracking tasks. However, since the number of parameters grows as the product of the input embedding and hidden dimension, their parameter counts can be extraordinarily large. While this may prove to be unproblematic in large-scale multi-task and language modeling tasks, data efficiency is a concern.

To gain insights on the data efficiency of bilinear models, we train and evaluate the models on the tasks discussed in the previous section, using fixed training set sizes. We also compare to LSTM. All models were trained on an input sequence length of 10, using the optimal learning rate found in the previous experiment. The results are shown in Figure 2. They show that despite the large number of parameters, the models are not less data efficient than the LSTM. This is true even of the full bilinear model (denoted "Bilinear" in the figure).

## 3.3 Multiplicative versus additive interactions

Our results on bilinear models are based solely on pure bilinear transformations, such that hidden units do not have any additive bias terms or any other additive dependencies on the inputs. Our results imply that such pure bilinear transformations are *sufficient* for learning state tracking tasks. In the following experiment, we study the performance of models with and without additive terms, and we show empirically that for some models, the absence of additive terms is also necessary (performance degrades in the presence of additive terms).

Table 1: In-distribution and length-generalization performance (normalized such that $0$ indicates random chance) of various models on three state tracking tasks: modular addition, simulating random state machines, and modular arithmetic. Bilinear models outperform others, with block-diagonal variants improving as block size increases. Non-linear recurrent models (LSTM, RNN) also perform well, whereas Mamba and Transformer struggle with long-sequence generalization.

| Modulus / State Size | | Validation Accuracy (Length 2-10) | | | | | | OOD Accuracy (Length 500) | | | | | |
|---|---|---|---|---|---|---|---|---|---|---|---|---|---|
| | | 2 | 3 | 5 | 10 | 25 | 50 | 2 | 3 | 5 | 10 | 25 | 50 |
| **Modular Addition** | | | | | | | | | | | | | |
| Bilinear | | 1.00 | 1.00 | 1.00 | 1.00 | 1.00 | 1.00 | 1.00 | 1.00 | 1.00 | 1.00 | 1.00 | 1.00 |
| Factored Bilinear | | 1.00 | 1.00 | 1.00 | 1.00 | 1.00 | 1.00 | 1.00 | 1.00 | 1.00 | 1.00 | 1.00 | 0.95 |
| Block Diag. (block size) | 1 | 1.00 | 0.96 | 0.88 | 0.85 | 0.45 | 0.32 | 1.00 | 0.00 | 0.00 | 0.10 | 0.00 | 0.02 |
| | 2 | 1.00 | 1.00 | 1.00 | 1.00 | 1.00 | 1.00 | 1.00 | 1.00 | 1.00 | 1.00 | 1.00 | 1.00 |
| | 8 | 1.00 | 1.00 | 1.00 | 1.00 | 1.00 | 1.00 | 1.00 | 1.00 | 1.00 | 1.00 | 1.00 | 1.00 |
| | 64 | 1.00 | 1.00 | 1.00 | 1.00 | 1.00 | 1.00 | 1.00 | 1.00 | 1.00 | 1.00 | 1.00 | 1.00 |
| $\mathcal{R}_2$ Block Diag. | | 1.00 | 1.00 | 1.00 | 1.00 | 1.00 | 1.00 | 1.00 | 0.00 | 1.00 | 0.66 | 0.37 | 0.00 |
| LSTM | | 1.00 | 1.00 | 1.00 | 1.00 | 1.00 | 0.99 | 1.00 | 1.00 | 0.98 | 1.00 | 0.00 | 0.02 |
| RNN | | 1.00 | 1.00 | 1.00 | 1.00 | 1.00 | 1.00 | 1.00 | 1.00 | 1.00 | 0.98 | 0.37 | 0.07 |
| Mamba (layers) | 1 | 0.99 | 0.92 | 0.96 | 0.85 | 0.74 | 0.61 | 0.00 | 0.01 | 0.01 | 0.00 | 0.00 | 0.00 |
| | 2 | 1.00 | 1.00 | 1.00 | 1.00 | 1.00 | 1.00 | 0.00 | 0.02 | 0.01 | 0.00 | 0.00 | 0.00 |
| | 4 | 1.00 | 1.00 | 1.00 | 1.00 | 1.00 | 0.47 | 0.01 | 0.01 | 0.00 | 0.01 | 0.00 | 0.00 |
| Transformer (layers) | 1 | 1.00 | 1.00 | 1.00 | 0.47 | 0.98 | 0.19 | 0.03 | 0.01 | 0.01 | 0.00 | 0.00 | 0.00 |
| | 2 | 1.00 | 1.00 | 1.00 | 0.99 | 0.89 | 0.00 | 0.01 | 0.02 | 0.00 | 0.00 | 0.00 | 0.00 |
| | 4 | 1.00 | 1.00 | 1.00 | 0.99 | 0.92 | 0.02 | 0.04 | 0.00 | 0.00 | 0.00 | 0.01 | 0.00 |
| **State Machine** | | | | | | | | | | | | | |
| Bilinear | | 1.00 | 1.00 | 1.00 | 1.00 | 1.00 | 1.00 | 1.00 | 1.00 | 1.00 | 1.00 | 1.00 | 1.00 |
| Factored Bilinear | | 1.00 | 1.00 | 1.00 | 1.00 | 1.00 | 0.19 | 1.00 | 1.00 | 1.00 | 1.00 | 1.00 | 0.01 |
| Block Diag. (block size) | 1 | 1.00 | 0.28 | 0.20 | 0.14 | 0.09 | 0.07 | 1.00 | 0.03 | 0.00 | 0.00 | 0.00 | 0.00 |
| | 2 | 1.00 | 1.00 | 0.84 | 0.49 | 0.25 | 0.15 | 1.00 | 0.34 | 0.16 | 0.06 | 0.06 | 0.02 |
| | 8 | 1.00 | 1.00 | 1.00 | 1.00 | 0.48 | 0.21 | 1.00 | 1.00 | 1.00 | 0.41 | 0.13 | 0.04 |
| | 64 | 1.00 | 1.00 | 1.00 | 1.00 | 1.00 | 1.00 | 1.00 | 1.00 | 1.00 | 1.00 | 1.00 | 1.00 |
| $\mathcal{R}_2$ Block Diag. | | 1.00 | 0.29 | 0.19 | 0.11 | 0.07 | 0.02 | 1.00 | 0.00 | 0.00 | 0.00 | 0.00 | 0.01 |
| LSTM | | 1.00 | 1.00 | 1.00 | 1.00 | 1.00 | 0.30 | 1.00 | 1.00 | 1.00 | 1.00 | 0.64 | 0.09 |
| RNN | | 1.00 | 1.00 | 1.00 | 1.00 | 0.41 | 0.18 | 1.00 | 1.00 | 1.00 | 0.99 | 0.19 | 0.07 |
| Mamba (layers) | 1 | 1.00 | 1.00 | 0.96 | 0.55 | 0.34 | 0.19 | 0.00 | 0.99 | 0.87 | 0.31 | 0.16 | 0.07 |
| | 2 | 1.00 | 1.00 | 1.00 | 0.79 | 0.44 | 0.30 | 0.00 | 1.00 | 0.96 | 0.42 | 0.18 | 0.09 |
| | 4 | 1.00 | 1.00 | 1.00 | 0.99 | 0.62 | 0.41 | 0.03 | 0.99 | 0.97 | 0.47 | 0.24 | 0.10 |
| Transformer (layers) | 1 | 1.00 | 0.94 | 0.83 | 0.46 | 0.27 | 0.18 | 0.03 | 0.01 | 0.02 | 0.01 | 0.00 | 0.00 |
| | 2 | 1.00 | 1.00 | 0.97 | 0.61 | 0.39 | 0.17 | 0.01 | 0.01 | 0.01 | 0.01 | 0.00 | 0.00 |
| | 4 | 1.00 | 1.00 | 1.00 | 0.84 | 0.49 | 0.17 | 0.00 | 0.02 | 0.01 | 0.00 | 0.00 | 0.00 |
| **Modular Arithmetic** | | | | | | | | | | | | | |
| Bilinear | | 1.00 | 1.00 | 1.00 | 1.00 | 1.00 | 1.00 | 1.00 | 1.00 | 1.00 | 1.00 | 1.00 | 0.99 |
| Factored Bilinear | | 1.00 | 0.34 | 0.90 | 0.09 | 0.03 | 0.03 | 1.00 | 0.24 | 0.37 | 0.06 | 0.04 | 0.03 |
| Block Diag. (block size) | 1 | 0.60 | 0.30 | 0.24 | 0.27 | 0.16 | 0.14 | 0.19 | 0.15 | 0.09 | 0.11 | 0.04 | 0.03 |
| | 2 | 0.99 | 0.77 | 0.53 | 0.52 | 0.27 | 0.21 | 0.37 | 0.00 | 0.08 | 0.12 | 0.03 | 0.05 |
| | 8 | 1.00 | 1.00 | 1.00 | 1.00 | 0.54 | 0.47 | 1.00 | 1.00 | 0.41 | 0.24 | 0.06 | 0.08 |
| | 64 | 1.00 | 1.00 | 1.00 | 1.00 | 1.00 | 0.66 | 1.00 | 1.00 | 1.00 | 1.00 | 0.40 | 0.21 |
| $\mathcal{R}_2$ Block Diag. | | 0.61 | 0.34 | 0.21 | 0.23 | 0.03 | 0.04 | 0.02 | 0.00 | 0.04 | 0.04 | 0.02 | 0.03 |
| LSTM | | 1.00 | 1.00 | 1.00 | 1.00 | 1.00 | 0.90 | 1.00 | 1.00 | 1.00 | 1.00 | 0.99 | 0.64 |
| RNN | | 1.00 | 1.00 | 1.00 | 1.00 | 0.82 | 0.22 | 1.00 | 1.00 | 1.00 | 1.00 | 0.55 | 0.16 |
| Mamba (layers) | 1 | 0.99 | 0.83 | 0.56 | 0.59 | 0.24 | 0.14 | 0.74 | 0.55 | 0.29 | 0.32 | 0.08 | 0.08 |
| | 2 | 1.00 | 0.99 | 0.80 | 0.93 | 0.35 | 0.33 | 0.85 | 0.38 | 0.41 | 0.29 | 0.11 | 0.07 |
| | 4 | 1.00 | 1.00 | 0.99 | 0.99 | 0.55 | 0.29 | 0.92 | 0.75 | 0.48 | 0.51 | 0.17 | 0.09 |
| Transformer (layers) | 1 | 0.88 | 0.63 | 0.46 | 0.32 | 0.10 | 0.08 | 0.19 | 0.02 | 0.04 | 0.01 | 0.03 | 0.01 |
| | 2 | 1.00 | 0.97 | 0.81 | 0.32 | 0.11 | 0.08 | 0.19 | 0.15 | 0.05 | 0.04 | 0.02 | 0.01 |
| | 4 | 1.00 | 0.99 | 0.75 | 0.32 | 0.07 | 0.08 | 0.19 | 0.12 | 0.03 | 0.03 | 0.02 | 0.03 |

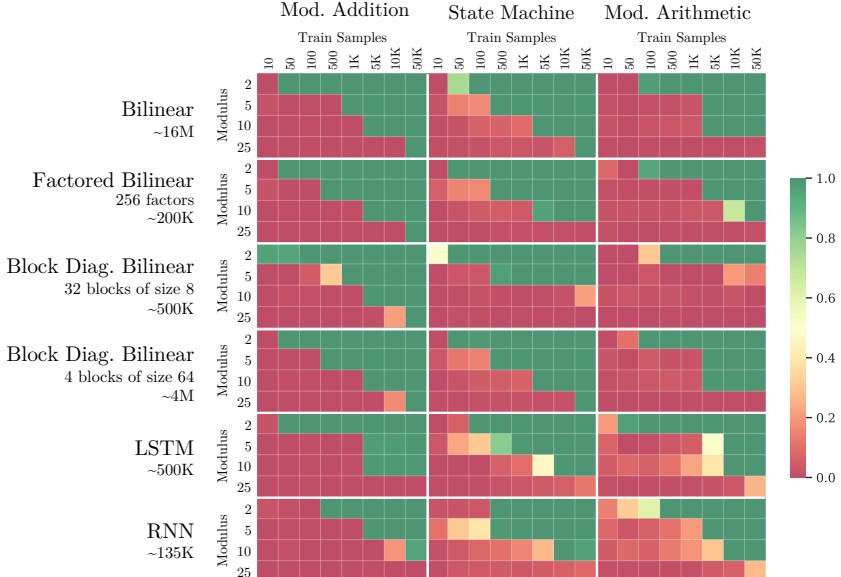

Figure 2: Data efficiency comparison between bilinear models and LSTM/RNN on state tracking tasks. All models were trained on sequences of length 10 and evaluated on length 500, with varying training set sizes. Despite their large parameter counts, bilinear models including the full variant, exhibit better data efficiency compared to LSTM.

Figure 3 (left) shows the OOD performance of real diagonal, 2D block-diagonal, and full bilinear models on the modular addition tasks discussed previously. As observed, and in line with the proposed hierarchy, the diagonal model is only capable of learning the parity task (only when the additive terms are excluded), and the 2D block-diagonal model can learn modular addition (again only without the additive terms). In contrast, additive contributions do not affect performance for the bilinear model. For a more comprehensive set of experiments and additional discussion on the effect of additive terms, refer to Appendix C.4.

### 3.4 Learning parity with a random network

Figure 3 (right) shows the OOD performance (testing length 400, best across 3 seeds and 2 learning rates) of this type of model, after training on sequences of lengths $10 - 50$. It shows, in line with the theoretical result, that the pure bilinear model can solve the task, even though recurrent parameters are frozen during training (only the readout layer is trained). It also shows the detrimental effect of additive terms for comparison.

Figure 3: (Left) Effect of (input-dependent) additive terms in the hidden state update rule on the OOD accuracy of modular addition task. (Right) Length generalization performance on parity with a random multiplicative RNN with and without additive terms.

| | Dataset | Parity | Modular Addition | | | |
|---|---|---|---|---|---|---|
| | Modulus | 2 | 3 | 5 | 10 | 25 |
| Model | Additive Terms | | | | | |
| Real Diag. | Yes | 0.00 | 0.00 | 0.01 | 0.00 | 0.00 |
| | No | 1.00 | 0.01 | 0.00 | 0.00 | 0.00 |
| 2D Block Diag. | Yes | 0.00 | 0.30 | 0.00 | 0.00 | 0.00 |
| | No | 1.00 | 1.00 | 1.00 | 1.00 | 0.98 |
| Bilinear | Yes | 1.00 | 1.00 | 1.00 | 1.00 | 1.00 |
| | No | 1.00 | 1.00 | 1.00 | 1.00 | 1.00 |

| Training Examples | | 2 | | | 100 | |
|---|---|---|---|---|---|---|
| Training Length | 10 | 20 | 50 | 10 | 20 | 50 |
| Additive Terms | | | | | | |
| Input Dependent | 0.05 | 0.07 | 0.05 | 0.03 | 0.05 | 0.04 |
| Input Dep. + Constant | 0.06 | 0.06 | 0.02 | 0.03 | 0.04 | 0.06 |
| Constant | 0.04 | 0.06 | 0.05 | 0.87 | 0.74 | 0.01 |
| None | 1.00 | 1.00 | 1.00 | 1.00 | 1.00 | 1.00 |

### 3.5 Representing commutative tasks by rotating phase angles

Figure 4 illustrates the rotation angles learned by the $\mathcal{R}_2$ block-diagonal model for the modular addition task with $m = 10$. The figure displays these angles for each input integer $(0, 1, \ldots, 9)$ across several 2-dimensional hidden state subspaces (12 out of 128). These subspaces are ordered based on the magnitude of the weights associated with these in the linear readout layer. The harmonic value, reported for each subspace, is calculated as $\theta_1/(2\pi/m)$, where $\theta_1$ is the learned rotation angle for the input integer 1 (rotation angles for other integers are multiples of $\theta_1$ in the representations learned by the model). This harmonic value indicates how closely $\theta_1$ aligns with an integer multiple of the fundamental angle $2\pi/m$ required for ideal cyclic group representation.

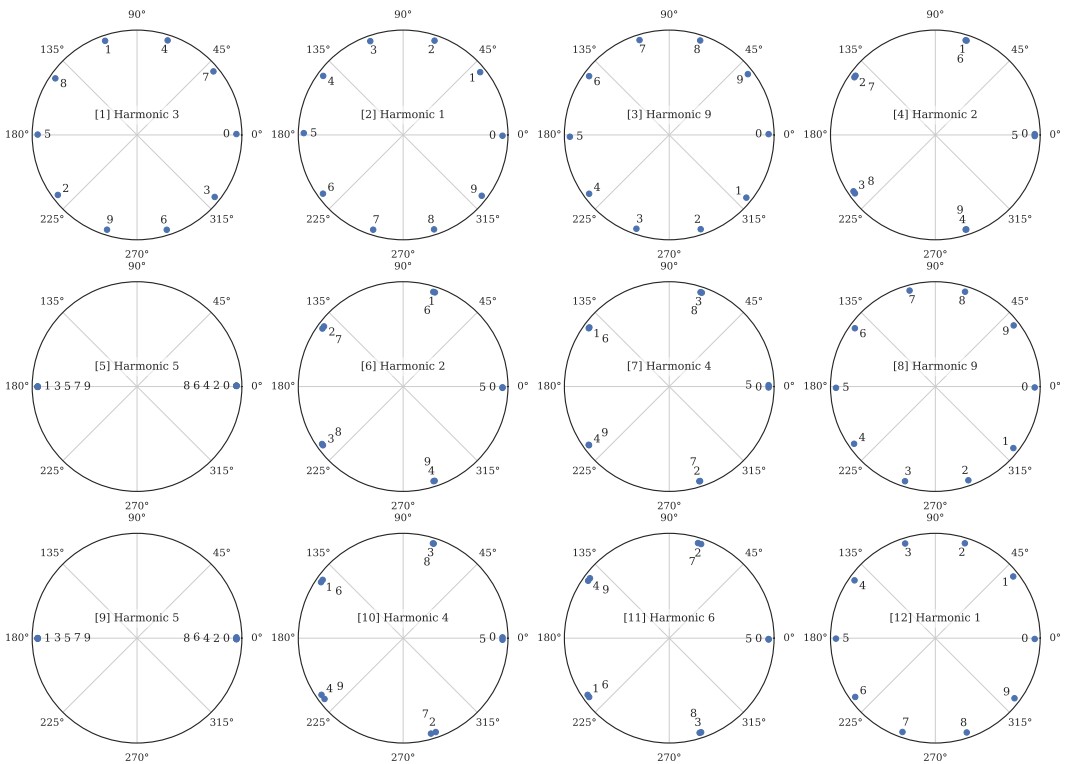

Figure 4: Visualization of the rotation angles learned by the $\mathcal{R}_2$ block-diagonal model for each input integer in the $m = 10$ modular addition task. Each subplot corresponds to a distinct 2-dimensional hidden state subspace. These subspaces are ordered based on the magnitude of the classifier weights. The "harmonic" is $\frac{\theta_1}{2\pi/m}$, where $\theta_1$ is the learned rotation angle for the integer 1.

## 4  Discussion

Our work shows that models in which hidden states depend bilinearly on previous hidden states and inputs can learn state tracking tasks. This is in contrast to many linear RNNs, such as Mamba (Gu and Dao, 2024), LRU (Orvieto et al., 2023), and others. It can also be viewed as extending upon the studies by Grazzi et al. (2025); Sarrof et al. (2024); Fan et al. (2024) to improve state tracking behavior beyond that work. However, it is important to note that bilinear models come at the cost of a parameter count that grows roughly cubically in the number of hidden states. Whether there are ways to reduce the number of parameters while retaining strong performance across a wide range of state tracking tasks is an important question for future research. A closely related question is whether such a reduction may be counterproductive (or conversely the large number of parameters even be beneficial) in massive multi-task scenarios such as language modeling, which is possible with linear models due to the possibility for efficient, parallel training.

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

# A Proofs

## A.1 Proof of Proposition 1

**Proposition 1.** *The bilinear state transition model defined in Equation* (2) *is capable of simulating any finite state machine* $\mathcal{S} = (Q, \Sigma, \delta, q_0)$.

*Proof.* Given any state machine $\mathcal{S} = (Q, \Sigma, \delta, q_0)$, and for all inputs $\sigma \in \Sigma$, we define the state transition matrix $\Delta_\sigma \in \{0,1\}^{|Q| \times |Q|}$ with the $(i,j)$-th element given by:

$$(\Delta_\sigma)_{ij} = \begin{cases} 1 & \text{if } \delta(q_j, \sigma) = q_i, \\ 0 & \text{otherwise.} \end{cases} \tag{6}$$

This means that if the state machine is in state $q_j$, and and receives input $\sigma$, the one-hot representation of the next state is precisely the $j$-th column of $\Delta_\sigma$. Consequently, if $q_t \in Q$ is the state at time $t$ and $h^t \in \{0,1\}^{|Q|}$ is its one-hot encoded representation, the state dynamics can be expressed as:

$$q_t = \delta(q_{t-1}, \sigma_t) \quad \Leftrightarrow \quad h^t = \Delta_{\sigma_t} h^{t-1} \tag{7}$$

To demonstrate that the bilinear state-transition form from Eq. (2) can represent any state machine, it suffices to show that the third-order tensor $\mathcal{W}$ can be constructed such that its resulting state transition matrix $\mathcal{A}_x$ equals $\Delta_\sigma$ for every $\sigma \in \Sigma$, where $x$ is the embedding corresponding to $\sigma$.

Without loss of generality, let the input alphabet be $\Sigma = \{1, 2, \ldots, |\Sigma|\}$. We define the input embedding $x \in \{0,1\}^{|\Sigma|}$ as the one-hot vector for the current input symbol $\sigma \in \Sigma$ (thus, the input dimension $D$ is effectively $|\Sigma|$). The third-order tensor $\mathcal{W}$ is then constructed such that for each $\sigma \in \Sigma$, the slice $\mathcal{W}_{..\sigma}$ is set equal to the state machine's transition matrix $\Delta_\sigma$. From Eq. (3), we will have:

$$(\mathcal{A}_x)_{ij} = \sum_k \mathcal{W}_{ijk} x_k = \mathcal{W}_{ij\sigma} = (\Delta_\sigma)_{ij}. \tag{8}$$

This is because $x$ is the one-hot vector for input symbol $\sigma$ (meaning $x_k = 1$ if $k = \sigma$, and $x_k = 0$ otherwise). Therefore, this construction yields $\mathcal{A}_x = \Delta_\sigma$. Consequently, the state-transition dynamics of the bilinear model, $h^t = \mathcal{A}_{x_t} h^{t-1}$, become equivalent to those of the state machine, $h^t = \Delta_{\sigma_t} h^{t-1}$.

$\square$

## A.2 Proof of Proposition 2

**Proposition 2.** *A linear RNN of the form in Eq.* (3) *with orthogonal state-transition matrices* $\mathcal{A}_x$ *that share a common, input-independent eigenbasis (i.e.,* $\mathcal{A}_x = \mathcal{P}\mathcal{D}_x\mathcal{P}^{-1}$ *with fixed* $\mathcal{P}$*) can simulate any abelian group (commutative operation).*

*Proof.* We show by construction that a bilinear model with state transition matrix that is block-diagonal with $\mathcal{R}_2$ rotation blocks can represent modular addition, and hence any cyclic group. Based on the fundamental theorem of finite abelian groups, every finite abelian group can be expressed as the direct sum of cyclic groups (typically of prime-power order) (Kurzweil and Stellmacher, 2004). Therefore, any model capable of simulating modular addition can, in principle, simulate any finite abelian group.

First, let's clarify how an orthogonal transition matrix simplifies to a block-diagonal form composed of 2-dimensional rotation blocks. (Further details are available in Section 2.3). Based on the Real Jordan Normal Form, an orthogonal matrix $\mathcal{A}_x$ is similar to a real block-diagonal matrix $\mathcal{D}_x$ (Axler, 2024). This means:

$$\mathcal{A}_x = \mathcal{P}_x \mathcal{D}_x \mathcal{P}_x^{-1}, \tag{9}$$

where $\mathcal{P}_x \in \mathbb{R}^{H \times H}$ is the invertible transformation matrix, and $\mathcal{D}_x \in \mathbb{R}^{H \times H}$ is a real block-diagonal matrix composed entirely of 2-dimensional rotation matrices (assuming $H$ is even), which we denote as

$$\mathcal{R}_2(\theta) = \begin{pmatrix} \cos\theta & -\sin\theta \\ \sin\theta & \cos\theta \end{pmatrix}. \tag{10}$$

Now, if the transformation matrix $\mathcal{P}_x$ is independent of the input $x$ (i.e., $\mathcal{P}_x = \mathcal{P}$), the fixed matrix $\mathcal{P}$ can be canceled out in the recurrence steps and absorbed into the input and output transformations of the recurrent layer. Therefore, for an orthogonal transition matrix with such input-independent transformations, we can effectively model $\mathcal{A}_x$ as being block-diagonal, with its blocks being 2-dimensional rotation matrices $\mathcal{R}_2(\theta(x))$, where the rotation angles $\theta(x)$ are parameterized by the input $x$.

Next, we present a simple construction with $H = 2$ (i.e., $\mathcal{A}_x = \mathcal{R}_2(\theta(x))$) that simulates modular addition. Let $x_\sigma$ be the embedding corresponding to an input integer $\sigma \in \mathcal{Z}_m = \{0, 1, \ldots, m-1\}$. We define the rotation angle for input $\sigma$ as $\theta(x_\sigma) = 2\pi\sigma/m$. Consequently, after observing a sequence of inputs $\sigma^1, \sigma^2, \ldots, \sigma^T$, the hidden state vector $h \in \mathbb{R}^2$ (with initial state $h^0$) evolves as follows:

$$h^T = \mathcal{A}_{x^T} h^{T-1} = \mathcal{A}_{x^1} \mathcal{A}_{x^2} \cdots \mathcal{A}_{x^T} h^0 \tag{11}$$

$$= \mathcal{R}_2\left(\theta\left(x^1\right)\right) \mathcal{R}_2\left(\theta\left(x^2\right)\right) \cdots \mathcal{R}_2\left(\theta\left(x^T\right)\right) h^0 \tag{12}$$

$$= \mathcal{R}_2\left(\sum_{t=1}^{T} \theta\left(x^t\right)\right) h^0 \tag{13}$$

$$= \mathcal{R}_2\left(\sum_{t=1}^{T} \frac{2\pi\sigma^t}{m}\right) h^0 \tag{14}$$

$\mathcal{R}_2\left(\phi\right) = \mathcal{R}_2\left(\phi \bmod 2\pi\right)$ $\qquad = \mathcal{R}_2\left(\left(\sum_{t=1}^{T} \frac{2\pi\sigma^t}{m}\right) \bmod 2\pi\right) h^0 \tag{15}$

$(2\pi\frac{\sigma}{m}) \bmod 2\pi = (\sigma \bmod m)(\frac{2\pi}{m})$ $\qquad = \mathcal{R}_2\left(\frac{2\pi}{m}\left(\sum_{t=1}^{T} \sigma^t \bmod m\right)\right) h^0 \tag{16}$

Let $y = \left(\sum_{t=1}^{T} \sigma^t\right) \bmod m$ be the target sum modulo $m$. This means $h^T$ is $h^0$ rotated by $\frac{2\pi}{m} y$:

$$h^T = \mathcal{R}_2\left(\frac{2\pi}{m} y\right) h^0. \tag{17}$$

Finally, since this rotation is unique for every possible value of $y \in \mathcal{Z}_m$, a linear readout layer (i.e., an $m$-class linear classifier) can perfectly extract $y$ from $h^T$:

$$y = \operatorname*{argmax}_{k \in \mathcal{Z}} w_k^\top h^T, \tag{18}$$

with

$$w_k = \mathcal{R}_2\left(\frac{2\pi}{m} k\right) h^0, \quad \forall k \in \mathcal{Z}_m. \tag{19}$$

$\square$

### A.3 Proof of Proposition 3

**Proposition 3.** *A random network with frozen real-diagonal transition matrix (without additive terms) and learnable linear readout layer learns parity with probability $1 - 2^{-H}$, for arbitrary sequence length from only 2 training examples of odd and even parity.*

*Proof.* For the parity task, the model observes an input sequence $\sigma^1, \sigma^2, \ldots, \sigma^T$, with each $\sigma^t \in \{0, 1\}$. The objective is to output the parity of this sequence, which is $\left(\sum_{t=1}^{T} \sigma^t\right) \bmod 2$.

Let $\mathcal{A}^{[0]}$ and $\mathcal{A}^{[1]}$ be the diagonal state-transition matrices corresponding to inputs 0 and 1, with $emba[0]$ and $emba[1]$ denoting the diagonal elements:

$$\mathcal{A}^{[0]} = \operatorname{diag}(a^{[0]}), \quad \mathcal{A}^{[1]} = \operatorname{diag}(a^{[1]}) \tag{20}$$

The hidden state evolves according to $h^t = \mathcal{A}^{[\sigma^t]} h^{t-1}$. For the $i$-th component of the hidden state, this evolution is:

$$h_i^T = a_i^{[\sigma^t]} h_i^{T-1} = h_i^0 \prod_{t=1}^{T} a_i^{[\sigma^t]}, \tag{21}$$

where $a_i^{(\sigma)}$ denotes the $i$-th diagonal element of $\mathcal{A}^{(\sigma)}$ (i.e., the $i$-th element of $a^{[\sigma]}$).

Crucially, the sign of the product $\prod_{t=1}^{T} a_i^{[\sigma^t]}$ can encode parity. If, for a given component $i$, we have $a_i^{[0]} > 0$ and $a_i^{[1]} < 0$, then $\text{sgn}\left(\prod_{t=1}^{T} a_i^{[\sigma^t]}\right)$ will be positive for an even number of 1s (even parity) and negative for an odd number of 1s (odd parity). This is because the number of negative terms ($a_i^{[1]}$) in the product matches the count of 1s in the input sequence. Conversely, if $a_i^{[0]} < 0$ and $a_i^{[1]} > 0$, the sign of the product becomes $(-1)^{T-\text{count of 1s}}$, which also encodes parity, albeit in a manner dependent on the sequence length $T$. Since $\text{sgn}\left(\prod_{t=1}^{T} a_i^{[\sigma^t]}\right) = \text{sgn}(h_i^T h_i^0)$, if $h_i^0$ is initialized with a fixed sign (e.g., positive), then in either case where $a_i^{[0]}$ and $a_i^{[1]}$ have opposite signs (i.e., $a_i^{[0]} a_i^{[1]} < 0$), the sign of $h_i^T$ contains sufficient information to determine the parity of the input sequence. The model then only needs to update its read-out layer (e.g., with one example of even and one of odd parity) to decode parity from $h_i^T$; all other recurrent parameters could remain fixed.

Assuming that the elements of $a^{(0)}$ and $a^{(1)}$ are i.i.d. and symmetrically distributed around 0 at initialization, we analyze the probability of finding such a suitable component $i$. The probability that an arbitrary component $i$ has $a_i^{[0]}$ and $a_i^{[1]}$ with opposite signs is 0.5. Therefore, given the independence across the $H$ components, the probability that there exists at least one component $i$ for which $a_i^{[0]} a_i^{[1]} < 0$ is $1 - (1 - 0.5)^H = 1 - 2^{-H}$.

$\square$

**Remark:** For the model to learn parity for arbitrary sequence lengths using a simple sign-based readout from a single component $i$, the ideal condition is $a_i^{[1]} < 0$ and $a_i^{[0]} > 0$. Under the same i.i.d. symmetric initialization assumptions, this specific configuration for a component $i$ occurs with probability $1/4$. Therefore, the probability that at least one such ideally suited component $i$ exists is $1 - (1 - 1/4)^H = 1 - (3/4)^H$.

# B Implementation details

## B.1 Tasks

For all tasks, to generate a training sample, we first randomly select the number of inputs $n \sim \mathcal{U}(2, N)$, where $N$ is the maximum training sequence length. We then select $n$ input symbols from $\{0, 1, \ldots, m-1\}$ uniformly at random with replacement. For the modular arithmetic task specifically, we also sample $n - 1$ operators uniformly with replacement from the set $\{+, -, \times\}$, which are then interleaved with the $n$ input symbols.

Each sample is structured with special tokens: it begins with a `[BOS]` (beginning of sequence) token and the input portion concludes with an `[EOI]` (end of input) token, immediately followed by the TARGET, as shown:

$$[BOS] \ INPUT_1 \ INPUT_2 \ INPUT_3 \ \cdots \ INPUT_n \ [EOI] \ TARGET$$

Each input symbols (including multi-digit integers and potentially operators), and special tokens `[BOS]` and `[BOI]` are tokenized as single tokens. During training and inference, all model outputs are disregarded except for the output corresponding to the `[EOI]` token; this output is taken as the model's prediction for TARGET. Consequently, during training, the loss is calculated only on this final target prediction.

We evaluate the models on the following three tasks:

**Modular addition:** The target is the sum of input integers modulo $m$. For example, with $n = 5$ and $m = 20$:

$$[BOS] \ 8 \ 0 \ 12 \ 18 \ 5 \ [EOI] \ 3$$

**Modular arithmetic:** This task involves processing a sequence alternating between $n$ integers from $\{0, 1, \ldots, m-1\}$ and $n - 1$ arithmetic operators from $\{+, \times, -\}$. The target is the result of these operations applied sequentially from left to right, with all calculations performed modulo $m$. For example, with $n = 5$ and $m = 20$:

$$[BOS] \ 3 \ * \ 9 \ - \ 17 \ + \ 6 \ + \ 12 \ [EOI] \ 8$$

The target is calculated as:

$$(3 \times 9) \bmod 20 = 7$$
$$(7 - 17) \bmod 20 = 10$$
$$(10 + 6) \bmod 20 = 16$$
$$(16 + 12) \bmod 20 = 8$$

**Simulating state machines:** The objective is to simulate a randomly generated finite state machine (FSM). Both the input alphabet $\Sigma$ and the set of states $Q$ are identical to $\{0, 1, \ldots, m-1\}$. For each state $q \in Q$, the transition function $\delta(q, \sigma)$ is defined as $\pi_q(\sigma)$, where $\pi_q$ is a random permutation of $\Sigma$. This transition function $\delta$ is generated once per FSM definition and remains fixed for all samples related to that FSM. The first input symbol in the sequence, $INPUT_1$, determines the initial state of the FSM. Subsequent symbols $INPUT_2, \ldots, INPUT_n$ are processed as inputs to the FSM, and the target is the FSM's final state. For example, consider $m = 6$ and the following randomly generated transition function:

|                    | **Input Symbol ($\sigma$)** | | | | | |
|--------------------|---|---|---|---|---|---|
|                    | 0 | 1 | 2 | 3 | 4 | 5 |
| **0** | 3 | 0 | 4 | 5 | 1 | 2 |
| **1** | 2 | 1 | 0 | 3 | 5 | 4 |
| **Current State ($q$)**    **2** | 5 | 0 | 2 | 1 | 3 | 4 |
| **3** | 5 | 0 | 1 | 2 | 4 | 3 |
| **4** | 1 | 0 | 3 | 4 | 2 | 5 |
| **5** | 5 | 4 | 0 | 3 | 1 | 2 |

An example sequence for this FSM would be:

```
[BOS] 4 1 2 5 5 [EOI] 2
```

The initial state is the first input (4 in this example), and upon observing each input, the state transitions occur based on the transition table:

$$\text{State: } 4, \text{ Input: } 1 \xrightarrow{\delta(4,1)=0} \text{New State: } 0$$

$$\text{State: } 0, \text{ Input: } 2 \xrightarrow{\delta(0,2)=4} \text{New State: } 4$$

$$\text{State: } 4, \text{ Input: } 5 \xrightarrow{\delta(4,5)=5} \text{New State: } 5$$

$$\text{State: } 5, \text{ Input: } 5 \xrightarrow{\delta(5,5)=2} \text{New State: } 2 \quad \text{(Target)}$$

## B.2 Experiment Setup

For the experiments discussed in Section 3.1 and reported in Table 1, all models were trained using the ADAM optimizer with three learning rates ($10^{-3}, 10^{-4}, 10^{-5}$), and the configuration yielding the best performance was selected for reporting. All models were trained from random initializations, without learning rate scheduling, weight decay, or dropout. In addition, the parameters of bilinear models (and variants) were initialized from a uniform distribution $\mathcal{U}(-0.01, 0.01)$. Training was conducted for 100,000 steps with a batch size of 64. An early stopping criterion was applied if the validation loss fell below $10^{-5}$. For these experiments, training examples were randomly sampled at each training step with input sequence lengths ranging from 2 to 10, while models were evaluated on inputs of length 500.

For the data efficiency experiments detailed in Section 3.2 (results in Figure 2), we used the optimal learning rate identified for each model and task from the previous experiment. We constructed fixed training sets of specified sizes and trained models for 1000 epochs over each set. Other settings were kept consistent with those in the previous experiment.

Regarding the baseline models in Table 1, the Transformer baseline is based on the GPT-2 architecture (Radford et al., 2019) with configurations of 1, 2, and 4 layers, and a model (embedding/hidden) dimension of 256, consistent with other models. Other parameters, such as an MLP inner expansion factor of 4, followed default GPT-2 (small) settings. We also used Mamba-1 (Gu and Dao, 2024) with 1, 2, and 4 layers, setting its model and hidden dimensions to 256. For other configurations, we adopted default values of Mamba-130M, including an intermediate expansion size of 512, a state space dimension of 16, and a convolution kernel size of 4.

All experiments were conducted on a cluster of A100 GPU nodes. A single training and evaluation run for a given model configuration, task, and setting typically completed on a single GPU within an hour in most cases, or up to a few hours in the worst case.

# C Additional experiments

## C.1 Parameter-matched models

The large parameter counts for some of our models (e.g., the full bilinear variant) are a direct result of matching hidden dimension (256) rather than parameters in experiments presented in Section 3. To address the concern about parameter efficiency, we conducted another set of experiments, similar to those in Table 1, in which models were matched in parameter count by adjusting their hidden dimension. In Table 2 we report the validation and OOD test accuracy on sequences of length 500, with training performed on sequences up to length 10. These new results still show superior state-tracking performance of bilinear models in most tasks (a slight degradation for modular arithmetic).

Table 2: Validation and OOD test accuracy for models matched by parameter count via adjusted hidden dimensions. Bilinear models maintain superior state-tracking performance across most tasks.

| Model | Spec. | Hidden Dim. | Layers | Parameters |
|---|---|---|---|---|
| Bilinear | | 80 | 1 | 513,207 |
| Factored Bilinear | Rank 700 | 256 | 1 | 541,447 |
| Block Diag. Bilinear | 4 Block of Size 32 | 128 | 1 | 526,215 |
| Block Diag. Bilinear | 32 Blocks of Size 8 | 256 | 1 | 528,135 |
| LSTM | | 256 | 1 | 529,927 |
| RNN | | 512 | 1 | 532,487 |
| Mamba | | 128 | 4 | 549,376 |
| Transformer | 4 Heads | 96 | 4 | 546,528 |

| | Validation Accuracy (Length 2-10) | | | | | | OOD Accuracy (Length 500) | | | | | |
|---|---|---|---|---|---|---|---|---|---|---|---|---|
| Modulus / State Size | 2 | 3 | 5 | 10 | 25 | 50 | 2 | 3 | 5 | 10 | 25 | 50 |
| **Modular Addition** | | | | | | | | | | | | |
| Bilinear | 1.00 | 1.00 | 1.00 | 1.00 | 1.00 | 1.00 | 1.00 | 1.00 | 1.00 | 1.00 | 1.00 | 1.00 |
| Factored Bil. (700 factors) | 1.00 | 1.00 | 1.00 | 1.00 | 1.00 | 1.00 | 1.00 | 1.00 | 1.00 | 1.00 | 1.00 | 1.00 |
| Block Diag. (Block Size 8) | 1.00 | 1.00 | 1.00 | 1.00 | 1.00 | 1.00 | 1.00 | 1.00 | 1.00 | 1.00 | 1.00 | 1.00 |
| Block Diag. (Block Size 32) | 1.00 | 1.00 | 1.00 | 1.00 | 1.00 | 1.00 | 1.00 | 1.00 | 1.00 | 1.00 | 1.00 | 1.00 |
| LSTM | 1.00 | 1.00 | 1.00 | 1.00 | 1.00 | 0.99 | 1.00 | 1.00 | 0.98 | 1.00 | 0.00 | 0.02 |
| RNN | 1.00 | 1.00 | 1.00 | 1.00 | 1.00 | 1.00 | 1.00 | 1.00 | 1.00 | 1.00 | 0.36 | 0.11 |
| Mamba | 1.00 | 1.00 | 1.00 | 1.00 | 1.00 | 0.98 | 0.01 | 0.00 | 0.01 | 0.01 | 0.00 | 0.00 |
| Transformer | 1.00 | 1.00 | 1.00 | 0.97 | 0.63 | 0.01 | 0.03 | 0.04 | 0.00 | 0.00 | 0.00 | 0.00 |
| **State Machine** | | | | | | | | | | | | |
| Bilinear | 1.00 | 1.00 | 1.00 | 1.00 | 1.00 | 1.00 | 1.00 | 1.00 | 1.00 | 1.00 | 1.00 | 1.00 |
| Factored Bil. (700 factors) | 1.00 | 1.00 | 1.00 | 1.00 | 1.00 | 0.98 | 1.00 | 1.00 | 1.00 | 1.00 | 1.00 | 0.18 |
| Block Diag. (Block Size 8) | 1.00 | 1.00 | 1.00 | 1.00 | 0.60 | 0.31 | 1.00 | 1.00 | 1.00 | 0.39 | 0.11 | 0.02 |
| Block Diag. (Block Size 32) | 1.00 | 1.00 | 1.00 | 1.00 | 1.00 | 0.73 | 1.00 | 1.00 | 1.00 | 1.00 | 1.00 | 0.11 |
| LSTM | 1.00 | 1.00 | 1.00 | 1.00 | 1.00 | 0.30 | 1.00 | 1.00 | 1.00 | 1.00 | 0.66 | 0.09 |
| RNN | 1.00 | 1.00 | 1.00 | 1.00 | 0.67 | 0.22 | 1.00 | 1.00 | 1.00 | 1.00 | 0.25 | 0.08 |
| Mamba | 1.00 | 1.00 | 1.00 | 0.99 | 0.61 | 0.33 | 0.00 | 1.00 | 0.96 | 0.48 | 0.24 | 0.08 |
| Transformer | 1.00 | 1.00 | 0.99 | 0.74 | 0.38 | 0.17 | 0.00 | 0.02 | 0.02 | 0.01 | 0.01 | 0.00 |
| **Modular Arithmetic** | | | | | | | | | | | | |
| Bilinear | 1.00 | 1.00 | 1.00 | 1.00 | 1.00 | 0.70 | 1.00 | 1.00 | 1.00 | 1.00 | 0.35 | 0.19 |
| Factored Bil. (700 factors) | 1.00 | 1.00 | 1.00 | 1.00 | 1.00 | 0.28 | 1.00 | 1.00 | 1.00 | 1.00 | 0.33 | 0.29 |
| Block Diag. (Block Size 8) | 1.00 | 1.00 | 1.00 | 1.00 | 0.66 | 0.60 | 1.00 | 1.00 | 0.12 | 0.20 | 0.03 | 0.07 |
| Block Diag. (Block Size 32) | 1.00 | 1.00 | 1.00 | 1.00 | 0.94 | 0.79 | 1.00 | 1.00 | 1.00 | 1.00 | 0.09 | 0.13 |
| LSTM | 1.00 | 1.00 | 1.00 | 1.00 | 1.00 | 0.90 | 1.00 | 1.00 | 1.00 | 1.00 | 0.94 | 0.64 |
| RNN | 1.00 | 1.00 | 1.00 | 1.00 | 1.00 | 0.35 | 1.00 | 1.00 | 1.00 | 1.00 | 0.98 | 0.29 |
| Mamba | 1.00 | 1.00 | 0.97 | 0.99 | 0.38 | 0.23 | 0.95 | 0.78 | 0.36 | 0.37 | 0.18 | 0.07 |
| Transformer | 1.00 | 0.58 | 0.26 | 0.25 | 0.07 | 0.09 | 0.19 | 0.04 | 0.01 | 0.02 | 0.01 | 0.01 |

## C.2 Simulating dihedral groups

Consider a finite state-machine with $Q = \{0, \cdots, m-1\} \times \{-1, +1\}$ and input alphabet $\Sigma = \{\texttt{advance}, \texttt{reverse}\}$. The state consists of a "value" and a binary "direction". Upon receiving input $\texttt{advance}$, the value will be incremented or decremented by 1 (modulo $m$) depending on the current direction. The input $\texttt{reverse}$ flips the direction while leaving the value unchanged. Formally, the transition function is defined as follows:

$$\delta((s, d), \texttt{advance}) = ((s + d) \bmod m, d)$$
$$\delta((s, d), \texttt{reverse}) = (s, -d)$$

See Example 6 in Liu et al. (2023) for more details on dihedral groups. Table 3 reports the in-distribution and length-generalization performance of various models when simulating dihedral groups with different moduli. As before, all models were trained on sequences of length 10 and evaluated on sequences of length 500.

Table 3: In-distribution and length-generalization performance of various models on simulating dihedral groups with different moduli.

| | Validation Accuracy (Length 2-10) | | | | | | OOD Accuracy (Length 500) | | | | | |
| --- | --- | --- | --- | --- | --- | --- | --- | --- | --- | --- | --- | --- |
| Modulus | 2 | 3 | 5 | 10 | 25 | 50 | 2 | 3 | 5 | 10 | 25 | 50 |
| Bilinear | 1.00 | 1.00 | 1.00 | 1.00 | 1.00 | 1.00 | 1.00 | 1.00 | 1.00 | 1.00 | 1.00 | 1.00 |
| Factored Bil. (256 factors) | 1.00 | 1.00 | 1.00 | 1.00 | 1.00 | 1.00 | 1.00 | 1.00 | 1.00 | 1.00 | 1.00 | 0.99 |
| Block Diag. (block size 64) | 1.00 | 1.00 | 1.00 | 1.00 | 1.00 | 1.00 | 1.00 | 1.00 | 1.00 | 1.00 | 1.00 | 1.00 |
| $\mathcal{R}_2$ Block Diag. | 0.78 | 0.32 | 0.35 | 0.42 | 0.40 | 0.41 | 0.02 | 0.03 | 0.00 | 0.07 | 0.01 | 0.01 |
| RNN | 1.00 | 1.00 | 1.00 | 1.00 | 1.00 | 1.00 | 1.00 | 1.00 | 1.00 | 1.00 | 1.00 | 0.07 |
| LSTM | 1.00 | 1.00 | 1.00 | 1.00 | 1.00 | 1.00 | 1.00 | 1.00 | 1.00 | 1.00 | 0.21 | 0.24 |
| Mamba | 1.00 | 1.00 | 1.00 | 1.00 | 1.00 | 1.00 | 0.15 | 0.01 | 0.00 | 0.00 | 0.00 | 0.00 |
| Transformer | 1.00 | 1.00 | 1.00 | 1.00 | 1.00 | 1.00 | 0.23 | 0.02 | 0.01 | 0.00 | 0.01 | 0.01 |

## C.3 Effect of number of factors

Similar to the results presented in Section 3.1, Table 4 presents the validation and out-of-distribution accuracy of factored bilinear models on the state machine simulation task, considering an increasing number of factors $(R)$ across various state space sizes $(m)$. As these results indicate, increasing the number of factors enables the simulation of larger state machines, as a factored model with a higher $R$ more closely approximates a full bilinear model.

Table 4: In-distribution and length-generalization (normalized) accuracy of factored bilinear models with different number of factors, on the state machine simulation task.

| | | Validation Accuracy (Length 2-10) | | | | | | OOD Accuracy (Length 500) | | | | | |
| --- | --- | --- | --- | --- | --- | --- | --- | --- | --- | --- | --- | --- | --- |
| | # States | 2 | 3 | 5 | 10 | 25 | 50 | 2 | 3 | 5 | 10 | 25 | 50 |
| Model | # Factors | | | | | | | | | | | | |
| | 1 | 0.00 | 0.02 | 0.00 | 0.02 | 0.01 | 0.00 | 0.03 | 0.00 | 0.00 | 0.00 | 0.01 | 0.00 |
| | 2 | 0.00 | 0.03 | 0.00 | 0.01 | 0.00 | 0.00 | 0.00 | 0.01 | 0.00 | 0.00 | 0.00 | 0.01 |
| | 4 | 1.00 | 1.00 | 0.65 | 0.02 | 0.01 | 0.00 | 1.00 | 0.97 | 0.29 | 0.01 | 0.00 | 0.00 |
| | 8 | 1.00 | 1.00 | 0.92 | 0.17 | 0.01 | 0.01 | 1.00 | 1.00 | 0.58 | 0.04 | 0.00 | 0.00 |
| | 16 | 1.00 | 1.00 | 1.00 | 0.44 | 0.06 | 0.01 | 0.95 | 1.00 | 1.00 | 0.26 | 0.01 | 0.01 |
| Factored Bilinear | 64 | 1.00 | 1.00 | 1.00 | 1.00 | 0.26 | 0.04 | 1.00 | 1.00 | 1.00 | 1.00 | 0.08 | 0.00 |
| | 128 | 1.00 | 1.00 | 1.00 | 1.00 | 0.68 | 0.10 | 1.00 | 1.00 | 1.00 | 1.00 | 0.26 | 0.00 |
| | 256 | 1.00 | 1.00 | 1.00 | 1.00 | 1.00 | 0.19 | 1.00 | 1.00 | 1.00 | 1.00 | 1.00 | 0.01 |
| | 512 | 1.00 | 1.00 | 1.00 | 1.00 | 1.00 | 0.45 | 1.00 | 1.00 | 1.00 | 1.00 | 1.00 | 0.10 |
| | 1024 | 1.00 | 1.00 | 1.00 | 1.00 | 1.00 | 1.00 | 1.00 | 1.00 | 1.00 | 1.00 | 1.00 | 0.78 |
| | 2048 | 1.00 | 1.00 | 1.00 | 1.00 | 1.00 | 1.00 | 1.00 | 1.00 | 1.00 | 1.00 | 1.00 | 1.00 |
| Bilinear | | 1.00 | 1.00 | 1.00 | 1.00 | 1.00 | 1.00 | 1.00 | 1.00 | 1.00 | 1.00 | 1.00 | 1.00 |

## C.4 Multiplicative vs. additive recurrence

In a model with bi-linear state transitions, the role of the inputs is to represent transformations on the hidden state, effectively acting as *computations* performed by hidden units. This contrasts with the conventional role of hidden units in an RNN, which primarily involves information retention. Therefore, for fully bilinear models, multiplicative interactions without additive contributions are sufficient for learning state transitions. In fact, for rotational and (block-)diagonal models, the inclusion of additive terms can even be detrimental.

We trained models under four configurations: with and without input-dependent additive contributions, with and without bias, and without any additive terms. Table 5 reports the OOD (sequence length 500) accuracy for the diagonal, 2D block-diagonal, and full bilinear models on parity, modular addition, and state machine simulation tasks.

As observed, and in line with the proposed hierarchy, the diagonal model is only capable of learning the parity task (only when the additive terms are excluded), the 2D block-diagonal model can learn modular addition (again only without the additive terms). In contrast, the full bilinear model is able to learn both modular addition and the non-commutative task, regardless of whether the additive terms are included.

Interestingly, in line with prior work by Terzić et al. (2025), we find that additive terms sometimes improve performance on tasks outside a model's hierarchy (e.g., the 2D block-diagonal model on non-commutative tasks), although overall accuracy remains low in such cases.

Table 5: Out-of-distribution (sequence length 500 for training sequence length of 10) accuracy of real diagonal, 2D block-diagonal, and full bilinear models on parity, modular addition, and state machine simulation tasks with and without additive terms in the recurrence.

| Dataset | Parity | Modular Addition | | | | | State Machine | | | | |
|---|---|---|---|---|---|---|---|---|---|---|---|
| Modulus/State Size | 2 | 3 | 5 | 10 | 25 | 2 | 3 | 5 | 10 | 25 |
| Model / Additive Terms | | | | | | | | | | | |
| Real Diag. — Const. | 0.00 | 0.02 | 0.00 | 0.00 | 0.00 | 0.13 | 1.00 | 0.57 | 0.19 | 0.11 |
| Real Diag. — Input Dependent | 0.00 | 0.01 | 0.00 | 0.00 | 0.00 | 0.03 | 1.00 | 0.75 | 0.24 | 0.11 |
| Real Diag. — Input Dep. + Const. | 0.00 | 0.00 | 0.01 | 0.00 | 0.00 | 0.00 | 1.00 | 0.74 | 0.24 | 0.11 |
| Real Diag. — None | 1.00 | 0.01 | 0.00 | 0.00 | 0.00 | 1.00 | 0.01 | 0.00 | 0.00 | 0.00 |
| 2D Block Diag. — Const. | 0.00 | 0.00 | 0.66 | 0.77 | 0.73 | 0.00 | 1.00 | 0.84 | 0.28 | 0.13 |
| 2D Block Diag. — Input Dependent | 0.00 | 0.00 | 0.02 | 0.00 | 0.00 | 0.04 | 1.00 | 0.95 | 0.32 | 0.12 |
| 2D Block Diag. — Input Dep. + Const. | 0.00 | 0.30 | 0.00 | 0.00 | 0.00 | 0.00 | 1.00 | 0.96 | 0.31 | 0.11 |
| 2D Block Diag. — None | 1.00 | 1.00 | 1.00 | 1.00 | 0.98 | 1.00 | 0.34 | 0.16 | 0.08 | 0.05 |
| Bilinear — Const. | 1.00 | 1.00 | 1.00 | 1.00 | 1.00 | 1.00 | 1.00 | 1.00 | 1.00 | 1.00 |
| Bilinear — Input Dependent | 1.00 | 1.00 | 1.00 | 1.00 | 1.00 | 1.00 | 1.00 | 1.00 | 1.00 | 1.00 |
| Bilinear — Input Dep. + Const. | 1.00 | 1.00 | 1.00 | 1.00 | 1.00 | 1.00 | 1.00 | 1.00 | 1.00 | 1.00 |
| Bilinear — None | 1.00 | 1.00 | 1.00 | 1.00 | 1.00 | 1.00 | 1.00 | 1.00 | 1.00 | 1.00 |

Our observation is consistent with work of Terzić et al. (2025), where it is shown that a complex diagonal model can generalize to longer sequences on a commutative automaton, but only when the additive term is removed and a linear readout is used. However, they also show the same model fails to learn a non-commutative automaton. In that case, the best performance is achieved when additive terms are included and a non-linear readout layer is used. Even then, generalization remains limited, although the additive terms offer a slight improvement.

