# OpenReview forum: "Revisiting Bi-Linear State Transitions in Recurrent Neural Networks"
_NeurIPS.cc/2025/Conference — NeurIPS 2025 poster_

### Official Review · Reviewer_9AkV · 2025-06-22

**Clarity:** 4
**Significance:** 4
**Originality:** 3
**Rating:** 5
**Confidence:** 4

**Summary:**

This paper analyzes bi-linear operations in RNNs (where hidden states and inputs interact multiplicatively) in the context of state tracking tasks. By parameterizing the hidden state matrices of SSMs in bilinear form, this theoretically show how to solve the state tracking task. The paper also introduces a hierarchy of task complexity, with simple linear RNNs (like Mamba) at the base.

**Questions:**

1. How do bi-linear RNNs compare to the other RNNs in Figure 1 on "real" sequential processing tasks? E.g., sequential MNIST?
2. Is there any connection to associative memory models, which also have multiplicative connections (albeit between hidden units, and not between inputs). For example, see Figure 1 [here](https://arxiv.org/pdf/2008.06996). There is also a (very) recent [paper](https://www.science.org/doi/epdf/10.1126/sciadv.adu6991) which contains multiplicative interactions between hidden states and external inputs. It would be interesting to know if there is some theoretical connection between bi-linear RNNs and associative memory.

**Ethical Concerns:**

["NO or VERY MINOR ethics concerns only"]

**Final Justification:**

The authors addressed all my (minor) comments. I maintain my recommendation to accept.

**Limitations:**

Yes.

**Paper Formatting Concerns:**

N/A.

**Quality:**

4

**Strengths And Weaknesses:**

The paper is very well-written, and the motivation is clear. I learned a lot reading this work!

One weakness is the lack of experiments on "real" benchmark sequential processing tasks, like those found in the [Long Range Arena](https://arxiv.org/abs/2011.04006).

---

> ### Author Rebuttal · Authors · 2025-07-31
>
> > 1. How do bi-linear RNNs compare to the other RNNs in Figure 1 on "real" sequential processing tasks? E.g., sequential MNIST/Long Range Arena?
>
> The primary focus of our work is to analyze the state-tracking capabilities of bilinear recurrent networks and identify the architectural principles that govern them. For this reason, we concentrated on synthetic tasks that allow for controlled and precise evaluation. That said, we agree that scaling these models for natural language pretraining is a crucial next step. We believe that robust state-tracking is a significant advantage for complex reasoning, but scaling up introduces challenges, including improving recall and managing gradient flow during parallelizable training on long context. These challenges could potentially be addressed by integrating gating mechanisms, specialized convolutions or memory layers, as done also for SSMs in the past. We'll add a discussion on this to the revised manuscript.
>
> ---
>
> > 2. Is there any connection to associative memory models, which also have multiplicative connections (albeit between hidden units, and not between inputs). For example, see Figure 1 here. There is also a (very) recent paper which contains multiplicative interactions between hidden states and external inputs. It would be interesting to know if there is some theoretical connection between bi-linear RNNs and associative memory.
>
> We thank the reviewer for this insightful question and for the pointers to the related work (which we will include in the paper). There is indeed a fundamental theoretical connection between the bilinear formulation of recurrence and associative memory networks, particularly with modern dense associative memory models (modern Hopfield networks). The foundational link is the principle of multiplicative interactions modulated by an external signal. While classic associative memories focused on hidden-to-hidden connections, modern Hopfield networks (variants with continuous state, or the IDP Hopfield model as pointed out by the reviewer) have an update rule reminiscent of attention mechanisms, where an input query dynamically weights the stored memory patterns. This is conceptually similar to the bilinear recurrence formulation, where the input modulates the hidden-state transition dynamics of the system in a multiplicative manner. We acknowledge that deeper understanding and study of this connection could lead to interesting future directions and appreciate the point raised by the reviewer.

---

> > ### Comment · Reviewer_9AkV · 2025-08-01
> > **All points addressed, thanks.**
> >
> > The authors have understood and addressed my points satisfactorily.
> >
> > I wish them good luck with this paper--I really enjoyed it!

---

### Official Review · Reviewer_Cy8L · 2025-06-28

**Clarity:** 4
**Significance:** 3
**Originality:** 3
**Rating:** 5
**Confidence:** 4

**Summary:**

In this paper, the authors study the properties of bilinear RNNs, where state transitions are governed by a tensor that is contracted against both the current state and the current input. They prove limitations on the state-tracking capabilities of several variants of standard SSMs and linear RNNs on state tracking tasks, and further prove that bilinear RNNs can simulate any finite state machine. Finally, they conduct a series of experiments on parity, modular arithmetic, and simulating random state machines to demonstrate that bilinear RNNs with more practical parameterizations (in terms of parameter efficiency) perform well on state tracking tasks when compared to other architectures.

**Questions:**

- Can the authors comment further on why including biases degrades performance of bilinear RNNs? Why can't the model learn to simply ignore the biases, and why does this lead to instabilities?

**Ethical Concerns:**

["NO or VERY MINOR ethics concerns only"]

**Final Justification:**

The additional experiments clarified concerns I had over fairly comparing different model classes in a parameter-matched way.

**Limitations:**

Yes

**Quality:**

4

**Strengths And Weaknesses:**

Strengths:

The paper is well-written, and to the best of my knowledge, clarifies important limits on the expressivity of various parameterizations of linear RNNs on state-tracking tasks. The analysis appears mathematically tight, and covers a wide range of interesting subproblems in state tracking (commutative vs noncommutative tasks, parity, automata).

Weaknesses:

While the authors do address the parameter-inefficient nature of bilinear RNNs by considering low-rank or factorized parameterizations, their experiments do not compare different model classes in a parameter-matched way; instead, hidden dimension is held fixed, giving a potentially massive parameter count advantage to the bilinear RNNs. A more complete set of experiments comparing bilinear RNNs and their factorized variants to other architectures with conserved parameter count would be needed to understand whether bilinear RNNs are actually practically useful on state-tracking tasks, and would be a useful addition to round out the paper regardless of what the results end up showing. I am open to bumping up my score to "Accept" if this can be done.

---

> ### Author Rebuttal · Authors · 2025-07-31
>
> > While the authors do address the parameter-inefficient nature of bilinear RNNs by considering low-rank or factorized parameterizations, their experiments do not compare different model classes in a parameter-matched way; instead, hidden dimension is held fixed, giving a potentially massive parameter count advantage to the bilinear RNNs. A more complete set of experiments comparing bilinear RNNs and their factorized variants to other architectures with conserved parameter count would be needed to understand whether bilinear RNNs are actually practically useful on state-tracking tasks, and would be a useful addition to round out the paper regardless of what the results end up showing. I am open to bumping up my score to "Accept" if this can be done.
>
> We thank the reviewer for their constructive suggestion. First, we would like to emphasize that we chose to use a consistent hidden dimension size of 256 for all models in Table 1 which resulted in very large parameter counts for some of the models.
> As suggested, we performed a new experiment to compare models in a parameter-matched setting instead, by adjusting the hidden state size of the models to approximately match parameter counts:
>
> | Model             |   Hidden Dim. |   Layers |   Parameters |
> |:------------------|-------------:|-------------:|-------------:|
> | Bilinear          |           80 |            1 |       513,207 |
> | Block Diag.  (Block Size = 32)   |          128 |            1 |       526,215 |
> | Factored Bil. (Rank = 700) |          256 |            1 |       541447 |
> | LSTM              |          256 |            1 |       529,927 |
> | RNN               |          512 |            1 |       532,487 |
> | Mamba             |          128 |            4 |       549,376 |
> | Transformer       |           96 |            4 |       546,528 |
>
> We report the OOD test accuracy on sequences of length 500, with training performed on sequences up to length 10. The results show the superior performance of bilinear models in most state-tracking tasks (but a mild degradation for modular arithmetic), even with matched parameter counts:
>
> | Model        |Mod. Add.|    2 |    3 |    5 |   10 |   25 |   50 |State Machine|  2   |  3   |  5   |  10  |  25  | 50   |Mod. Arith.|  2   |  3   |  5   | 10   | 25   | 50   |
> |:-------------|---------|:----:|------| -----|------|------|------|-------------|------|------|------|------| -----| ---- |-----------|------|------|------|------|------|------|
> | Bilinear     |         | 1.00 | 1.00 | 1.00 | 1.00 | 1.00 | 1.00 |             | 1.00 | 1.00 | 1.00 | 1.00 | 1.00 | 1.00 |           | 1.00 | 1.00 | 1.00 | 1.00 | 0.35 | 0.19 |
> | Block Diag.  |         | 1.00 | 1.00 | 1.00 | 1.00 | 1.00 | 1.00 |             | 1.00 | 1.00 | 1.00 | 1.00 | 1.00 | 0.11 |           | 1.00 | 1.00 | 1.00 | 1.00 | 0.09 | 0.13 |
> | Factored Bil.|         | 1.00 | 1.00 | 1.00 | 1.00 | 1.00 | 1.00 |             | 1.00 | 1.00 | 1.00 | 1.00 | 1.00 | 0.18 |           | 1.00 | 1.00 | 1.00 | 1.00 | 0.33 | 0.29 |
> | RNN          |         | 1.00 | 1.00 | 1.00 | 1.00 | 0.36 | 0.11 |             | 1.00 | 1.00 | 1.00 | 1.00 | 0.25 | 0.08 |           | 1.00 | 1.00 | 1.00 | 1.00 | 0.98 | 0.29 |
> | LSTM         |         | 1.00 | 1.00 | 0.98 | 1.00 | 0.00 | 0.02 |             | 1.00 | 1.00 | 1.00 | 1.00 | 0.66 | 0.09 |           | 1.00 | 1.00 | 1.00 | 1.00 | 0.94 | 0.64 |
> | Mamba        |         | 0.01 | 0.00 | 0.01 | 0.01 | 0.00 | 0.00 |             | 0.00 | 1.00 | 0.96 | 0.48 | 0.24 | 0.08 |           | 0.95 | 0.78 | 0.36 | 0.37 | 0.18 | 0.07 |
> | Transformer  |         | 0.03 | 0.04 | 0.00 | 0.00 | 0.00 | 0.00 |             | 0.00 | 0.02 | 0.02 | 0.01 | 0.01 | 0.00 |           | 0.19 | 0.04 | 0.01 | 0.02 | 0.01 | 0.01 |
>
> Thank you again for the suggestion. We will add these results to the revised version of the paper.
>
> ---
>
> >  Can the authors comment further on why including biases degrades performance of bilinear RNNs? Why can't the model learn to simply ignore the biases, and why does this lead to instabilities?
>
> Our hypothesis is that the presence of additive terms may alter the loss landscape such that, while a zero-bias solution exists, it may not be stable or easily reachable via gradient descent. This is similar to how different initialization schemes can lead optimization toward different local minima, even when the networks can represent better solutions. We would also like to highlight an additional benefit of removing bias terms from the recurrence: the model becomes scale-invariant, which enables normalization of the hidden states to maintain stability over long sequences. This is demonstrated in our length generalization setup with sequences of length 500.

---

> > ### Comment · Reviewer_Cy8L · 2025-08-04
> >
> > The authors have addressed my concerns regarding the experiments satisfactorily with additional experiments. As such, I am increasing my score to 5.

---

### Official Review · Reviewer_RHHC · 2025-06-29

**Clarity:** 3
**Significance:** 3
**Originality:** 2
**Rating:** 3
**Confidence:** 5

**Summary:**

This paper focuses on bi-linear RNNs where the transition matrix is a bi-linear function (a multiplicative interaction) of current input and hidden activations of the previous time-step. The paper showed that bi-linear RNNs are highly effective at learning state tracking tasks by mainly removing any additive components, s.t. the hidden state is a true bi-linear (not affine function) of the previous time-step hidden state and input. This leads to a scale-invariance which allows for keeping hidden activations stable during training and inference.

**Questions:**

Q1) The empirical evaluations in Table 1 are limited to only three synthetic tasks. How about the non-commutative automata with solvable transformation semigroups, such as the dihedral group? A pointer for these tasks: B. Liu, et al. "Transformers Learn Shortcuts to Automata", ICLR 2023.

Q2) In Table 1, what is the reason for the drop in the OOD accuracy of RNNs as the Module increases? RNNs have sufficient expressiveness to learn any regular language.

Q3) The best-performing model is reported (out of three using different learning rates). It is also good to report the min-max, or average-max, performance across models.

Q4) Table 2 (right) shows the detrimental effect of additive terms on the parity automata, which is consistent with the prior art. How about benchmarking when learning a non-commutative automaton (e.g., D_30)? Could the model provide better length generalization once the B matrix is introduced? See Table 4 of the prior art.

Q5) Is there any sample efficiency comparison with RNNs in Figure 2?

Q6) What are the potential limitations and implications of applying this method towards making LLMs on real-world datasets?

**Ethical Concerns:**

["NO or VERY MINOR ethics concerns only"]

**Final Justification:**

After carefully considering the authors' responses as well as the discussion among the other reviewers, I have decided to maintain my score of 3. My reasoning is as follows:

1) The work shares substantial theoretical, conceptual, and empirical similarities with prior art (SD-SSM). Both approaches investigate the state-tracking abilities of linear RNNs, advocate for unstructured dense transition matrices, and highlight the limitations of complex diagonal matrices in modeling non-commutative operations. In effect, SD-SSM is a parameter-efficient case of the bilinear model proposed in this work, so it remains unclear whether the proposed bilinear model could have any practical edge over SD-SSM in the studied state-tracking.

1.1) It seems that the bilinear model (that does not inherently include additive terms) is an unstable system without input accumulation, whereas SD-SSM was stable (if p<1) with input accumulation.

2) As also noted by other reviewers, it is not yet evident how the improvements in state-tracking translate to downstream performance on real-world datasets. This is in contrast with other structured alternatives like block-diagonal linear RNNs, which showed competitive OOD generalization on the state-tracking as well as achieving superior results in real-world classification tasks e.g., on multivariate time-series: https://arxiv.org/pdf/2505.17761

**Limitations:**

What are the potential limitations and implications of applying this method towards making LLMs on real-world datasets?

**Paper Formatting Concerns:**

No formatting issue.

An extremely small minor typo: Add a full stop in Line 301: "using fixed training set sizes[.] We also compare"

**Quality:**

3

**Strengths And Weaknesses:**

Strengths:

- Bi-linear RNNs showed OOD generalization on synthetic state-tracking tasks
- The models show similar sample efficiency to LSTM

Weaknesses:

Several theoretical, conceptual, and experimental similarities with prior art (A. Terzic, et al., "On the Expressiveness and Length Generalization of Selective State-Space Models on Regular Languages", AAAI 2025) have not been mentioned in the paper. These should be clarified and elaborated:

> We show that linear RNNs with diagonal transition matrices are a special case limited to learning state tracking tasks with commutative structure. This restriction is true even for complex-valued diagonal transition matrices. Hence, linear RNNs with block-diagonal transition matrices of size 2 × 2 are not able to learn general state machines (negative result).

Theoretically, it has also been shown in the prior art that under a described mapping of FSA to a single-layer selective (i.e., input-dependent) state-space model (SSM) whose transition matrices are simultaneously diagonalizable, the selective SSM can only emulate commutative automata.

Moreover, the prior art had already noted that models that do not utilize the B matrix (i.e., additive interactions) tend to learn solutions that exhibit better length generalization than their counterparts (which include the B matrix).

Conceptually, the proposed state transition matrix A is fully parametrized through a linear transformation of the input x. In the prior art, the A matrix is a convex combination of a set of transition matrices based on the current input.

Experimentally, a single recurrent cell is followed by a linear classification head over the hidden states. This is also similar to what has been used in the prior art, with further ablation that shows the inefficiency of non-linear readouts.

---

> ### Author Rebuttal · Authors · 2025-07-31
>
> > Several theoretical, conceptual, and experimental similarities with prior art...
>
> Thank you for pointing us to the work by Terzic et al. (2025). It is a well-written paper with interesting connections to our work. We will add a detailed discussion in the related work section and highlight similarities and differences in other relevant subsections.
> We acknowledge the similarities: both papers analyze the state-tracking capabilities of linear RNNs, advocate for dense transitions to improve state-tracking, and show that complex diagonal transition matrices can only learn commutative operations. However, our work also differs in several key aspects.
> The model in Terzic et al. constructs its transition matrix as a convex combination of a fixed set of learnable dense matrices. This can be viewed as a specific, parameter-efficient approximation of a more general bilinear model, where the transition matrix is fully parameterized by the input. Our approach to stability also differs: while they use additive terms in the recurrence and normalize the columns of the transition matrix, our bilinear recurrence formulation does not inherently include additive terms. This enables scale-invariance and allows for direct normalization of the hidden states at arbitrary points in time.
> Furthermore, our work contributes a more detailed hierarchy of state-tracking by showing, both theoretically and experimentally, that a real-diagonal model can only learn parity, but with extreme data efficiency. Finally, we introduce and study two parameter-efficient variants: the factorized and block-diagonal bilinear models.
>
> ---
>
> > Q1) The empirical evaluations in Table 1 are limited to only three synthetic tasks. How about the non-commutative automata with solvable transformation semigroups, such as the dihedral group? A pointer for these tasks: B. Liu, et al.
>
> We thank the reviewer for this suggestion and the pointer. Following the suggestion, we benchmarked all models on the dihedral group simulation task in the same setup as discussed in the paper (training length 10, testing length 500, best of 3 learning rates). As the results below demonstrate, bilinear RNN models successfully learn a length-generalizable solution to this task:
>
> | Model  |Valid.|2|3|5|10|25|50|Test|2|3|5|10|25|50|
> |:--|--|--|--|--|--|--|--|--|--|--|--|--|--|--|
> |Bilinear  ||1.00|1.00|1.00|1.00|1.00|1.00||1.00|1.00|1.00|1.00|1.00|1.00|
> |Block Diag.  ||1.00|1.00|1.00|1.00|1.00|1.00||1.00|1.00|1.00|1.00|1.00|1.00|
> |Factored Bil.||1.00|1.00|1.00|1.00|1.00|1.00||1.00|1.00|1.00|1.00|1.00|0.99|
> |2D Rotation  ||0.78|0.32|0.35|0.42|0.40|0.41||0.02|0.03|0.00|0.07|0.01|0.01|
> |LSTM||1.00|1.00|1.00|1.00|1.00|1.00||1.00|1.00|1.00|1.00|0.21|0.24|
> |RNN ||1.00|1.00|1.00|1.00|1.00|1.00||1.00|1.00|1.00|1.00|1.00|0.07|
> |Mamba  ||1.00|1.00|1.00|1.00|1.00|1.00||0.15|0.01|0.00|0.00|0.00|0.00|
> |Transformer  ||1.00|1.00|1.00|1.00|1.00|1.00||0.23|0.02|0.01|0.00|0.01|0.01|
>
> ---
>
> > Q2) In Table 1, what is the reason for the drop in the OOD accuracy of RNNs as the Module increases? RNNs have sufficient expressiveness to learn any regular language.
>
> We hypothesize that the observed drop in accuracy could be due to the learning problem becoming harder as the modulus (m) grows. A larger modulus means a larger state space and a more complex transition function for the model to learn. This could make the optimization more difficult and amplify the well-known challenges of propagating gradients through time in standard RNNs. Therefore, while the required solution is representable within the RNN's capacity, the model may struggle to converge to it as the task complexity increases.
>
> ---
>
> > Q3) The best-performing model is reported (out of three using different learning rates). It is also good to report the min-max, or average-max, performance across models.
>
> We thank the reviewer for this suggestion. We will add a table to the appendix showing the minimum and maximum accuracy for each model across the learning rates, complementing the best-performing results presented in the main text.
>
> ---
>
> > Q4) Table 2 (right) shows the detrimental effect of additive terms on the parity automata, which is consistent with the prior art. How about benchmarking when learning a non-commutative automaton (e.g., D_30)? Could the model provide better length generalization once the B matrix is introduced? See Table 4 of the prior art.
>
> The mentioned work shows that the complex diagonal model can generalize to longer sequences on a commutative automaton, but only when the additive term is removed and a linear readout is used. However, the same model fails to learn a non-commutative automaton. In that case, the best performance is achieved when additive terms are included and a non-linear readout layer is used. Even then, generalization remains limited, though the additive terms offer a slight improvement.
>
> We performed an ablation study on the non-commutative state machine simulation task. We report results for the diagonal, 2D block-diagonal, and full bilinear models on parity, modular addition, and state machine simulation task (as a reminder, please note that accuracy numbers are scaled so that 0.0 represents random chance performance):
>
> **Validation Accuracy (Length 2-10)**
>
> Model|AdditiveTerms|Parity | Mod. Add. |5|10|25| State Machine |5|10|25|
> ---|:--:|:--:|:--:|:---:|:---:|:---:|:--:|:---:|:---:|:--:|
> RealDiagonal|Yes|1.00||0.54|0.03|0.01||0.83|0.36|0.11|
> RealDiagonal|No|1.00||0.88|0.85|0.40||0.22|0.15|0.09|
> 2D-BlockDiagonal|Yes|1.00||1.00|1.00|0.01||1.00|0.52|0.12|
> 2D-BlockDiagonal|No|1.00||1.00|1.00|1.00||0.83|0.49|0.23|
> Bilinear|Yes|1.00||1.00|1.00|1.00||1.00|1.00|1.00|
> Bilinear|No|1.00||1.00|1.00|1.00||1.00|1.00|1.00|
>
> **Test Accuracy (Length 500)**
>
> Model|AdditiveTerms|Parity|Mod. Add. |5|10|25| State Machine |5|10|25|
> --|:--:|:--:|:--:|:--:|:--:|:--:|:--:|:--:|:--:|:--:|
> RealDiagonal|Yes|0.00||0.01|0.00|0.00||0.74|0.24|0.11|
> RealDiagonal|No|1.00||0.00|0.00|0.00||0.00|0.00|0.00|
> 2D-BlockDiagonal|Yes|0.00||0.00|0.00|0.00||0.96|0.31|0.11|
> 2D-BlockDiagonal|No|1.00||1.00|1.00|0.98||0.16|0.08|0.05|
> Bilinear|Yes|1.00||1.00|1.00|1.00||1.00|1.00|1.00|
> Bilinear|No|1.00||1.00|1.00|1.00||1.00|1.00|1.00|
>
> As observed, and in line with the proposed hierarchy, the diagonal model is only capable of learning the parity task (only when the additive terms are excluded), the 2D block-diagonal model can learn modular addition (again only without the additive terms). In contrast, the full bilinear model is able to learn both modular addition and the non-commutative task, regardless of whether the additive terms are included. Consistent with prior work, we find that additive terms improve the models' performance on the non-commutative task, although the overall performance remains low in both cases. We will include this discussion in the revised version of the paper, along with citations to the relevant work.
>
> ---
>
> > Q5) Is there any sample efficiency comparison with RNNs in Figure 2?
>
> Based on your question, we also performed the data-efficiency experiments for the RNN using the same setup described in Section 3.2, and we report the results below:
>
> **Modular Addition**
>
> Modulo|10|50|100|500|1000|5000|10000|50000|
> --|---:|---:|---:|---:|--:|--:|--:|--:|
> 2|0.03|0.01|0.00|1.00|1.00|1.00|1.00|1.00|
> 5|0.00|0.01|0.01|0.00|0.00|1.00|1.00|1.00|
> 10|0.01|0.01|0.00|0.00|0.00|0.00|0.19|0.95|
> 25|0.00|0.00|0.00|0.00|0.00|0.01|0.00|0.00|
>
> **State Machine**
>
> States|10|50|100|500|1000|5000|10000|50000|
> --|--:|---:|---:|--:|--:|--:|--:|--:|
> 2|0.02|0.02|0.04|1.00|1.00|1.00|1.00|1.00|
> 5|0.11|0.31|0.39|0.98|1.00|1.00|1.00|1.00|
> 10|0.01|0.04|0.08|0.11|0.16|0.27|1.00|0.96|
> 25|0.01|0.03|0.04|0.05|0.04|0.05|0.06|0.09|
>
> **Modular Arithmetic**
>
> Modulo|10|50.|100|500|1000|5000|10000|50000|
> --|--:|--:|---:|--:|--:|--:|--:|--:|
> 2|0.14|0.32|0.61|1.00|1.00|1.00|1.00|1.00|
> 5|0.08|0.03|0.06|0.11|0.20|1.00|1.00|1.00|
> 10|0.05|0.08|0.07|0.13|0.19|0.30|1.00|1.00|
> 25|0.01|0.01|0.01|0.01|0.02|0.05|0.06|0.28|
>
> Compared to the LSTM, we found that the simple RNN shows slightly lower data efficiency on the modular addition and state-machine tasks, but slightly better efficiency on the modular arithmetic task when using larger moduli. We will include this in Figure 2.
>
> ---
>
> > Q6) What are the potential limitations and implications of applying this method towards making LLMs on real-world datasets?
>
> The benefits of improved state-tracking are particularly evident in math and code benchmarks. There is supporting evidence in the literature, for example, Grazzi et al. (2025) (figure 10), which shows that enabling even the weakest form of state-tracking in models like Mamba and DeltaNet leads to improved perplexity in math and code. Therefore, we believe that models with stronger state-tracking capabilities will have a significant advantage in reasoning more generally.
>
> Currently, almost all large-scale pretrained language models have limited internal state-tracking capabilities. Although they rely on chain of thought to make the state observable in the token space, this becomes impractical when the state is large or continuous.
>
> Although bilinear models demonstrate strong state-tracking, we identify two potential limitations for scaling them to large-scale pretraining on natural language. First is their recall performance, which is crucial in the natural language domain. A second potential challenge is managing gradient flow over very long sequences while maintaining parallelizable training. Similar to recent developments in SSMs, limitations like these could be overcome by utilizing additional gating mechanisms, skip connections, 1D convolutions, or even hybrid layers that incorporate memory modules. While this is out of scope for this work, we do consider it important future work.
>
> _Grazzi, Riccardo, et al. "Unlocking State-Tracking in Linear RNNs Through Negative Eigenvalues." The Thirteenth International Conference on Learning Representations._

---

> > ### Comment · Reviewer_RHHC · 2025-08-03
> >
> > Thank you for the additional clarifications and the new results, particularly on the non-commutative tasks. After carefully considering the authors' responses as well as the discussion among the other reviewers, I have decided to maintain my score of 3. My reasoning is as follows:
> >
> > 1) The work shares substantial theoretical, conceptual, and empirical similarities with prior art (SD-SSM). Both approaches investigate the state-tracking abilities of linear RNNs, advocate for unstructured dense transition matrices, and highlight the limitations of complex diagonal matrices in modeling non-commutative operations. In effect, SD-SSM is a parameter-efficient case of the bilinear model proposed in this work, so it remains unclear whether the proposed bilinear model could have any practical edge over SD-SSM in the studied state-tracking.
> >
> > 1.1) It seems that the bilinear model (that does not inherently include additive terms) is an unstable system without input accumulation, whereas SD-SSM was stable (if p<1) with input accumulation.
> >
> > 2) As also noted by other reviewers, it is not yet evident how the improvements in state-tracking translate to downstream performance on real-world datasets. This is in contrast with other structured alternatives like block-diagonal linear RNNs, which showed competitive OOD generalization on the state-tracking as well as achieving superior results in real-world classification tasks e.g., on multivariate time-series: https://arxiv.org/pdf/2505.17761

---

> > > ### Author Response · Authors · 2025-08-05
> > >
> > > We thank the reviewer for their feedback, discussion, and suggestions for the additional experiments.
> > >
> > > > theoretical, conceptual, and empirical similarities with prior art (SD-SSM).
> > >
> > > We respectfully disagree with this point. We consider SD-SSM to be in the same line of work as our paper, namely assessing and improving the state-tracking capabilities of parallelizable recurrent neural networks. While we agree there are interesting connections as part of a growing interest in this area, we believe our work is distinct from and complementary to that work. We would like to clarify the key differences:
> > >
> > > - The bilinear recurrence in our work is defined as $h_i^t = \text{Bilinear}(h^{t-1}, x^t), \ \forall i$. In contrast, SD-SSM builds upon the standard linear RNN form $h^t =  A(x^t)h^{t-1} + b(x^t)$, with a specific structure for $A$. Our formulation excludes additive terms and we show both theoretically and empirically that they are not necessary for state-tracking tasks. Moreover, we show how the absence of additive terms allows us to stabilize training via normalization of hidden states.
> > >
> > > - We introduce two parameter-efficient approximations to the bilinear recurrence and evaluate their performance. We also demonstrate that different bilinear variants correspond to tasks of varying complexity: models with 2D block-diagonal transition matrices can solve commutative tasks (Terzic et al. have similar results for the special case of complex diagonal models), while models with real diagonal matrices can learn parity extremely efficiently (not considered by Terzic et al.).
> > >
> > > - The empirical results differ in fundamental ways. While the tasks are mostly distinct, we systematically increase the state size ($m$) of our tasks to evaluate performance as a function of task difficulty, aided by aforementioned stabilization. We also evaluate data efficiency of different types of model across all tasks. Furthermore, additional empirical results, as suggested by the reviewer, are now included, showing the viability of bilinear models which, unlike Terzic et al., are shown for variable state size (up to $m=50$).
> > >
> > > We welcome the growing interest in this area. Connections between works are natural; for instance, the work cited by the reviewer shares similarities with Fan et al. (2024), including their normalization and parallelization approach (in contrast to our approach, which is distinct from both).
> > >
> > >
> > > > 1.1) It seems that the bilinear model (that does not inherently include additive terms) is an unstable system without input accumulation, whereas SD-SSM was stable (if p<1) with input accumulation.
> > >
> > > SD-SSM ensures stability by normalizing the columns of the transition matrix, while we achieve stability by normalizing the hidden states instead. This is evident from our test setup, where the model remains stable at a test sequence length of 500, despite being trained only up to length 10.
> > >
> > > > 2) how the improvements in state-tracking translate to downstream performance on real-world datasets…
> > >
> > > Our primary focus is to analyze the state-tracking capabilities of bilinear recurrent networks and identify the architectural principles that govern them. To this end, we use synthetic tasks that allow for controlled and precise evaluation, as common in this line of work. We agree that scaling these models to real-world tasks, such as natural language pretraining is an important next step, however, we consider it to be out of scope for the current study.

---

### Official Review · Reviewer_LWAc · 2025-07-01

**Clarity:** 4
**Significance:** 2
**Originality:** 2
**Rating:** 4
**Confidence:** 3

**Summary:**

This paper investigates the expressivity of bi-linear recurrent models theoretically and empirically by studying their performance on three formal language tasks: modular addition, random state machine, and modular arithmetic.
The paper investigates different factorizations and parameterizations of the state transition tensor.
In the experiments bi-linear recurrent neural networks perform best on the state tracking tasks (with LSTM being close).

**Questions:**

- What are the exact parameter counts of the models? Since it is mentioned in the discussion section that the parameter count is larger due to the cubic scaling with embedding dim, it would make this point more clear if the parameter counts are added to Table 1, too.
- L 271: Why do you need to normalize during inference and not during training?

**Ethical Concerns:**

["NO or VERY MINOR ethics concerns only"]

**Final Justification:**

The authors clarified my questions and concerns regarding the parameter counts and the real world experiments during the rebuttal.
While the authors agree that applications to language pretraining would be a crucial next step, the paper focuses on theoretical justification and synthetic experiments, which seems reasonable.
Therefore, I will maintain my score.

**Limitations:**

yes

**Paper Formatting Concerns:**

No concerns.

**Quality:**

3

**Strengths And Weaknesses:**

Strengths:
- Very well written, clear and easy to understand
- Table 1: Good performance of Bilinear models (even though LSTM is close)

Weaknesses:
- The paper mentions that the stability problems of bi-linear recurrent models were the main reason for them not being used, and seems to solve this issue by dropping any additive terms in the state update. However, further (details on) experiments on this issue are missing.
- There are no „real world“ experiments, e.g. on language or vision. It could be at least explained if these models are applicable to a more large scale training setup?

---

> ### Author Rebuttal · Authors · 2025-07-31
>
> > The paper mentions that the stability problems of bi-linear recurrent models were the main reason for them not being used, and seems to solve this issue by dropping any additive terms in the state update. However, further (details on) experiments on this issue are missing.
>
> Yes, one benefit of removing additive terms from the recurrence is that the model becomes scale-invariant, which allows for normalization of the hidden states to ensure stability. The evidence for this stability is our strong length generalization result: models trained on sequences up to length 10 performed successfully on sequences of length 500, which is made possible by performing iterative normalization during inference. We thank the reviewer for raising this point; we will clarify the link between scale-invariance, normalization, and our OOD results in the updated version of the paper.
>
> ---
>
> > There are no „real world“ experiments, e.g. on language or vision. It could be at least explained if these models are applicable to a more large scale training setup?
>
> The primary focus of our work is to analyze the state-tracking capabilities of bilinear recurrent networks and identify the architectural principles that govern them. For this reason, we concentrated on synthetic tasks that allow for controlled and precise evaluation. That said, we agree that scaling these models for natural language pretraining is a crucial next step. This would be possible in principle, because unlike non-linear RNNs (but like linear RNNs/SSMs) all relevant computations could be parallelized for bilinear RNNs. We believe that robust state-tracking is a significant advantage for complex reasoning, but scaling up introduces challenges, including improving recall and managing gradient flow during parallelizable training on long context. These challenges can be effectively addressed by integrating gating mechanisms, specialized convolutions or memory layers. We'll add a discussion on these future directions to the revised manuscript.
>
> ---
>
> > What are the exact parameter counts of the models? Since it is mentioned in the discussion section that the parameter count is larger due to the cubic scaling with embedding dim, it would make this point more clear if the parameter counts are added to Table 1, too.
>
> Thank you for this helpful suggestion. While this information is already provided for a subset of models in Figure 2, we will add the parameter counts additionally for all models to Table 1 for added clarity:
>
> - Full Bilinear: ~16.7M
> - Block Diagonal, block size 1: ~69K, block size 2: ~135K, block size 8: ~528K, block size 64: ~ 4.2M
> - Factored Bilinear rank 1: ~4K, rank 2: ~5K, rank 4: ~7K, rank 8: ~10K, rank 16: ~16K, rank 64: ~53K, rank 128: ~102K, rank 256: ~200K, rank 512: ~397K, rank 1024: ~790K, rank 2048: ~1.5M
> - 2D Rotation: ~38K
> - LSTM: 531K
> - RNN: 135K
> - Transformer 1-layer: ~1M, 2-layer: ~1.8M, 4-layer: ~3.4M
> - Mamba 1-layer: 473K, 2-layer: 943K, 4-layer: 1.8M
>
> However, we also want to clarify a key point regarding these numbers: the large parameter counts for some of our models (e.g., the full bilinear variant) are a direct result of matching hidden dimension (256) rather than parameters in these experiments. To address the valid concern about parameter efficiency, we conducted a new set of experiments where models are matched in parameter count by adjusting their hidden dimension:
>
> | Model             |   Hidden Dim. |   Layers |   Parameters |
> |:------------------|-------------:|-------------:|-------------:|
> | Bilinear          |           80 |            1 |       513,207 |
> | Block Diag.  (Block Size = 32)   |          128 |            1 |       526,215 |
> | Factored Bil. (Rank = 700) |          256 |            1 |       541447 |
> | LSTM              |          256 |            1 |       529,927 |
> | RNN               |          512 |            1 |       532,487 |
> | Mamba             |          128 |            4 |       549,376 |
> | Transformer       |           96 |            4 |       546,528 |
>
> We report the OOD test accuracy on sequences of length 500, with training performed on sequences up to length 10. These new results still show superior state-tracking performance of bilinear models in most tasks (a slight degradation for modular arithmetic). We will add them to the updated version of the paper.
>
>
> | Model        |Mod. Add.|    2 |    3 |    5 |   10 |   25 |   50 |State Machine|  2   |  3   |  5   |  10  |  25  | 50   |Mod. Arith.|  2   |  3   |  5   | 10   | 25   | 50   |
> |:-------------|---------|:----:|------| -----|------|------|------|-------------|------|------|------|------| -----| ---- |-----------|------|------|------|------|------|------|
> | Bilinear     |         | 1.00 | 1.00 | 1.00 | 1.00 | 1.00 | 1.00 |             | 1.00 | 1.00 | 1.00 | 1.00 | 1.00 | 1.00 |           | 1.00 | 1.00 | 1.00 | 1.00 | 0.35 | 0.19 |
> | Block Diag.  |         | 1.00 | 1.00 | 1.00 | 1.00 | 1.00 | 1.00 |             | 1.00 | 1.00 | 1.00 | 1.00 | 1.00 | 0.11 |           | 1.00 | 1.00 | 1.00 | 1.00 | 0.09 | 0.13 |
> | Factored Bil.|         | 1.00 | 1.00 | 1.00 | 1.00 | 1.00 | 1.00 |             | 1.00 | 1.00 | 1.00 | 1.00 | 1.00 | 0.18 |           | 1.00 | 1.00 | 1.00 | 1.00 | 0.33 | 0.29 |
> | RNN          |         | 1.00 | 1.00 | 1.00 | 1.00 | 0.36 | 0.11 |             | 1.00 | 1.00 | 1.00 | 1.00 | 0.25 | 0.08 |           | 1.00 | 1.00 | 1.00 | 1.00 | 0.98 | 0.29 |
> | LSTM         |         | 1.00 | 1.00 | 0.98 | 1.00 | 0.00 | 0.02 |             | 1.00 | 1.00 | 1.00 | 1.00 | 0.66 | 0.09 |           | 1.00 | 1.00 | 1.00 | 1.00 | 0.94 | 0.64 |
> | Mamba        |         | 0.01 | 0.00 | 0.01 | 0.01 | 0.00 | 0.00 |             | 0.00 | 1.00 | 0.96 | 0.48 | 0.24 | 0.08 |           | 0.95 | 0.78 | 0.36 | 0.37 | 0.18 | 0.07 |
> | Transformer  |         | 0.03 | 0.04 | 0.00 | 0.00 | 0.00 | 0.00 |             | 0.00 | 0.02 | 0.02 | 0.01 | 0.01 | 0.00 |           | 0.19 | 0.04 | 0.01 | 0.02 | 0.01 | 0.01 |
>
> ---
>
> > L 271: Why do you need to normalize during inference and not during training?
>
> We did not apply normalization during training simply because the input sequences were short enough. It is entirely possible to apply hidden state normalization during both training and inference, and we will clarify in the revised text that its omission during our training runs was for simplicity, not a requirement of the model.

---

> > ### Author Response · Authors · 2025-08-07
> >
> > We again thank the reviewer for their thoughtful feedback. We have made our best effort to address all the questions and concerns raised, and we hope our responses reflect that.
> > If there is anything further we could clarify or improve to help the reviewer consider adjusting their score, we would be grateful for the opportunity to do so.

---

### Comment · Area_Chair_SyBF · 2025-08-01
**Reviewer-author discussion period**

Dear reviewers,

We are now in the reviewer-author discussion period until Aug 6 11:59pm AoE. The authors have posted a detailed response to each review. At your earliest convenience, please read all the reviews and rebuttals, and respond as soon as you can to the author's rebuttals to start discussion. At minimum, please respond to the author's rebuttal to indicate you have read it. The discussion period is not very long, so it would be good to ensure there is time for back-and-forth if needed.

Thanks for all your efforts!

Best, AC

---

### Decision · Program_Chairs · 2025-09-17

**Decision:**

Accept (poster)

**Comment:**

Strengths
- Proposed models show sample efficiency on synthetic state-tracking

- Reviewers note the paper is very-well written and easy to understand.

- The proposed bilinear models have good performance (though LSTM is close), and demonstrating OOD performance.

- Theoretically sound and justified, e.g. "tight mathematical analysis, covering a wide range of interesting subproblems in state tracking" (Reviewer Cy8L)

Weaknesses
- Substantial technical overlap with SD-SSM ([A. Terzic, et al., "On the Expressiveness and Length Generalization of Selective State-Space Models on Regular Languages", AAAI 2025] noted by Reviewer RHHC), and initial version of the paper did not include reference or discussion of this; after rebuttal, authors have committed to include reference and detailed discussion of this method relative to their work.

- Lack of parameter-matched evaluations (Reviewer Cy8L); rebuttal resolved this by providing evaluation on this.

- Lack of experimentation on "real" benchmark sequential processing tasks e.g. Long Range Arena (Reviewer 9AkV, Reviewer LWAc). Authors note they use synthetic tasks to more easily analyze and control state-tracking task, which makes sense. Authors satisfied this reviewer's other minor comments in rebuttal.

Decision

Majority of reviewers feel positively about this paper. Most critical reviewer is concerned about too much overlap with the prior work SD-SSM, but I think authors justify that there are differences with this method, and authors promise to cite and discuss relation to this work in the final revision.